# Nanopore blockade sensors for ultrasensitive detection of proteins in complex biological samples

Kyloon Chuah[1,4], Yanfang Wu[1,4], S.R.C. Vivekchand[1], Katharina Gaus[2], Peter J. Reece[3], Adam P. Micolich[3] & J. Justin Gooding [1]

Nanopore sensors detect individual species passing through a nanoscale pore. This experimental paradigm suffers from long analysis times at low analyte concentration and non-specific signals in complex media. These limit effectiveness of nanopore sensors for quantitative analysis. Here, we address these challenges using antibody-modified magnetic nanoparticles ((anti-PSA)-MNPs) that diffuse at zero magnetic field to capture the analyte, prostate-specific antigen (PSA). The (anti-PSA)-MNPs are magnetically driven to block an array of nanopores rather than translocate through the nanopore. Specificity is obtained by modifying nanopores with anti-PSA antibodies such that PSA molecules captured by (anti-PSA)-MNPs form an immunosandwich in the nanopore. Reversing the magnetic field removes (anti-PSA)-MNPs that have not captured PSA, limiting non-specific effects. The combined features allow detecting PSA in whole blood with a 0.8 fM detection limit. Our 'magnetic nanoparticle, nanopore blockade' concept points towards a strategy to improving nanopore biosensors for quantitative analysis of various protein and nucleic acid species.

---

[1] School of Chemistry, Australian Centre for NanoMedicine and the ARC Centre of Excellence in Convergent Bio-Nano Science and Technology, The University of New South Wales, Sydney, NSW 2052, Australia. [2] EMBL Australia Node in Single Molecule Science, School of Medical Sciences and the ARC Centre of Excellence in Advanced Molecular Imaging, The University of New South Wales, Sydney, NSW 2052, Australia. [3] School of Physics, The University of New South Wales, Sydney, NSW 2052, Australia. [4]These authors contributed equally: K. Chuah, Y. Wu. Correspondence and requests for materials should be addressed to J.J.G. (email: justin.gooding@unsw.edu.au)

Nanopore sensors are one of the first class of single-molecule sensors employed for quantitative analysis[1]. Single molecule sensitivity is achieved by detecting the transient decrease in ionic conductivity through the nanopore caused when a molecule of interest translocates and thus partially blocks the nanopore[1]. Quantitative analysis involves counting the transient conductivity events as molecules pass sequentially through the pore[2]. There are two major challenges that arise for quantitative analysis using nanopore sensors. Firstly, the requirement for analyte molecules to diffuse to near the pore to translocate one at a time means the time interval between detections at the femtomolar concentrations required for measuring rare species, e.g., for cancer diagnosis[3], is of order minutes[4]. Thus quantitative analysis at such low concentrations takes many (>10) hours[5,6]. Secondly, as any species that can translocate through the nanopore can give a signal, selectivity in complex biological fluids, e.g., blood, is exceedingly challenging. The difficulty is that to detect individual protein molecules, the pore must be a few nanometers in diameter. Any protein translocating the pore will thus generate a strong current transient regardless of its identity. That is, the lack of selectivity is intrinsic to the nanopore sensor design. As a result, while nanopores have seen great success in molecular identification, such as DNA sequencing, their progress towards quantitative analysis[7] has been impeded by these challenges. Solving the time and selectivity challenges would open the path to a new generation of nanopore-based biomolecular quantitative analysis systems[6,8,9]. These would provide single molecule resolution, could be calibration-free as they rely on counting alone[10], are compatible with bulk manufacture[11], and require simple electronics that are easily miniaturised for point-of-care and mobile use[12–17].

Attempts to decrease nanopore detection limits from the micromolar level have focused on extending the strong electric field generated near the nanopore further out into the bulk solution with significant success. The idea is to electrophoretically attract the analyte towards the pore. For example, Meller et al.[9] lowered the electrolyte concentration on the cis-side of the pore (0.2 M) relative to the trans-side (4 M). This gave a 30-fold increase in translocation rate and improved the detection limit to 3.8 pM. Freedman et al.[8] improved performance further using dielectrophoretic trapping, achieving detection limits of 5 fM with 315 events per minute. As such strategies are not compatible with complex biological fluids, there is scope for new approaches to push towards and below 1 fM in detection limit in complex biological fluids.

Our solution involves using antibody-labelled magnetic nanoparticles to capture and shuttle analytes to an antibody-labelled nanopore, exploiting the forces generated by an externally applied magnetic field to reduce analysis time. Once brought to the nanopore, the magnetic nanoparticles cannot translocate through the nanopore and blocks it instead. If the analyte is captured by the magnetic nanoparticle, the magnetic nanoparticles forms a sandwich-complex in the nanopore such that it cannot be removed when the namgnetic field is reversed to pull the nanoparticle out of the nanopore. In this way false signals are avoided and hence better specificity is achieved. This is a significant change in the operational paradigm—rather than passively waiting for the analyte to find and translocate through the nanopore by diffusion, we actively draw analyte to the nanopore by external field effect. This departure from the traditional nanopore sensor approach also provides significantly easier fabrication as much larger nanopores (~130 nm on the cis side and 30 nm on the trans side) can be used. Magnetic nanoparticles (MNPs) have long been used for pre-concentrating analyte prior to detection[18]. This enables sensors to achieve exquisitely low detection limits[19].

We use anti-PSA-labelled magnetic nanoparticles ((anti-PSA)-MNPs) dispersed into the sample to selectively capture a protein analyte, prostate-specific antigen (PSA). PSA is used as a model analyte here. The idea can be generalised to other proteins for which there are suitable antibodies. The case for detecting PSA at low concentration is for better treatment efficacy[20]. As such there has been a number of microdevices developed for detecting ultralow levels of PSA[21–25]. The (anti-PSA)-MNPs are brought rapidly to the nanopore under an applied magnetic field. The nanopore is modified with anti-PSA antibodies for a different PSA epitope. If an (anti-PSA)-MNP has captured the analyte, it will form a sandwich complex with the antibodies in the nanopore. This leads to long-term nanopore blockade and a permanent step-wise decrease in current (Fig. 1). The magnetic field direction is then reversed to draw any (anti-PSA)-MNPs that have not captured PSA, away from the nanopore to avoid false counts. We show this approach allows lower detection limits than previously reported for nanopore sensors (sub-fM), as well as analyte specificity. Furthermore, the blockade design means nanopores much larger than proteins can be employed such that the nanopore blockade sensor can be used in whole blood, as shown below, without proteins that translocate through the pore giving significant resistive spikes.

## Results

**Construction of nanopore blockade sensor.** We used solid-state nanopores formed in silicon nitride (SiN) with electron beam lithography (EBL) by the method shown in Supplementary Fig. 1. The pore diameter is controlled by the electron beam dose (Supplementary Figs. 2 and 3). The nanopores have a truncated cone structure (Fig. 1a) with an estimated sidewall angle of $70 \pm 2°$ (mean $\pm$ s.d., $n = 10$), consistent with the EBL nanopore fabrication method used by Wei et al.[26]. The truncated conical geometry is believed to be caused by a combination of the anisotropic etching of SiN by $CF_4$ plasma and the undercut profile of the developed electron beam resist due to electron back-scattering[27]. As a result, the SiN around the pore was not completely protected from reactive ion etching (RIE) process and partial etching of the SiN occurs in that region (Fig. 1a(i)). Single nanopores and nanopore arrays were fabricated. In this paper only single nanopores and $3 \times 3$ arrays are explored (Fig. 1a). The overall ionic current as a function of the number of nanopores in the array (Supplementary Fig. 4) and the power spectral density as a function of applied potential (Supplementary Fig. 5) were characterised.

The cis-side of the nanopore was modified with a self-assembled monolayer composed of a silane proximal moiety bonded to the SiN, an hexa(ethylene oxide) unit to resist non-specific protein adsorption and a distal moiety to which an antibody can be attached (Fig. 1a(ii))[28]. Evidence for the modification (the schemes for the surfaces present in Supplementary Table 1) was obtained from X-ray photoelectron spectroscopy (Supplementary Fig. 6) and the change in ionic current (Supplementary Fig. 7) with the electronic noise remaining unaffected (Supplementary Fig. 8). The change in the ionic current suggests a reduction in the nanopore diameter of ~5.7 nm, consistent with the surface of the nanopore being modified with the silane-$EG_6$ species of length ~2.77 nm. Antibodies recognising epitope 1 on PSA were immobilised onto the nanopore interiors[29]. The conical nanopore profile, with cis diameter of ~130 nm and trans diameter of ~30 nm, was crucial for the nanoparticle blockade. It allows the nanoparticles to enter the nanopore but not to translocate through it. The cubic magnetic nanoparticles were 50 nm in diameter (Supplementary Fig. 10) and prepared using methods described previously[30]. The

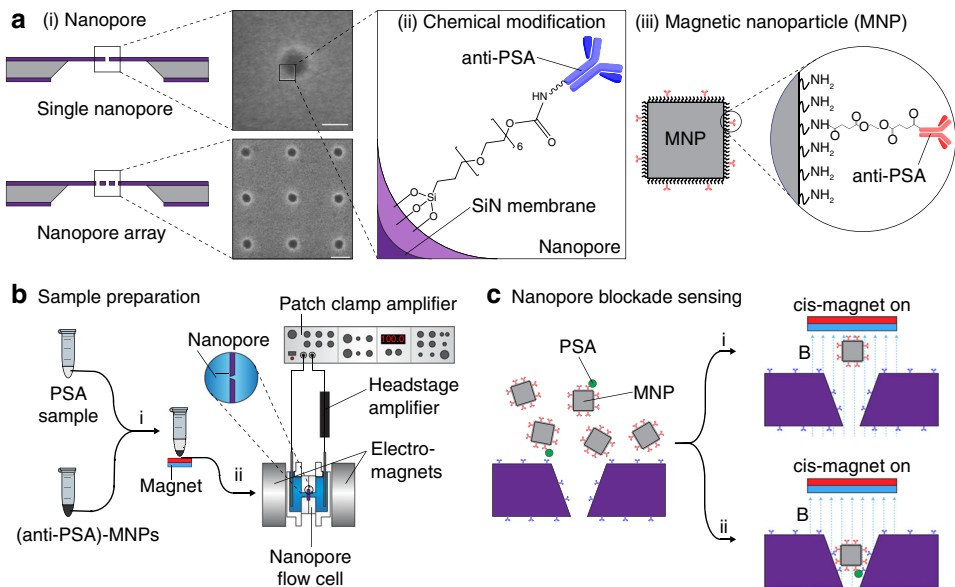

**Fig. 1** Nanopore blockade sensor. **a** (i) Schematics and scanning electron micrographs of (top) a solid-state nanopore, inset: a 48 nm nanopore, scale bar: 100 nm; (bottom) a nanopore array in SiN membrane, inset: a 3 × 3 array of nanopores with 0.5 μm spacing between each pores, scale bar: 200 nm. (ii) Illustration of a chemically modified SiN nanopore with silane-$EG_6$-(anti-PSA) self-assembled monolayer (SAM); EG equates to ethylene glycol. (iii) Illustration of an (anti-PSA)-conjugated MNP used in this work. Inset: chemical structure of the (anti-PSA)-$EG_6$ immobilised onto the amine-rich PEI coating of the MNP. **b** Flow chart illustrates the sample preparation steps for detection of PSA in a sample. (i) Sample containing PSA was mixed with (anti-PSA)-MNPs. After exposing the MNPs to the sample, the MNPs were then magnetically separated and washed 3 times with PBS. (ii) The MNPs were subsequently added into the cis-chamber of a nanopore flow cell filled with KCl as electrolyte. A potential difference was applied between both sides of the nanopore. **c** Active filtering of non-specific nanopore blockade events can be achieved by applying a magnetic field from the cis-side. (i) (Anti-PSA)-MNP that non-specifically adsorbed onto the pore surface will be ejected and thus unblocks the pore, whereas, (ii) (anti-PSA)-MNP that contains captured PSA will continue blocking the pore. Blue dashed lines represent the magnetic field lines, B

magnetic nanoparticles were coated in polyethylene imine to which a second anti-PSA antibody, recognising epitope 5 on PSA, was added to give the (anti-PSA)-MNPs (Fig. 1a(iii)).

A patch clamp setup to measure nanopore blockade events was used, with the fluid cell sandwiched between two electromagnets. The electromagnets were configured such that only one electromagnet is active at any time (Fig. 1b). The switching has no significant influence on ionic current (Supplementary Fig. 9). The trans-magnet was switched on to draw the (anti-PSA)-MNP towards the nanopore followed by switching to the cis-magnet to draw the (anti-PSA)-MNP away from the pore (Fig. 1c). A SEM image of a MNP inside the nanopore is shown in Supplementary Fig. 11. The cell volume was 700 μL with $7.4 \times 10^7$ MNPs on the cis side of the nanopore. The typical steady-state ionic current I was ~2 nA for a single 27 nm nanopore. To increase the probability of MNPs being captured in a nanopore, the direction of the applied magnetic field was alternated until a decrease in ionic current was detected, indicating that a MNP had successfully entered the nanopore. Recording the ionic current after the trans-magnet was switched on, we observed a slow decrease followed by a drastic drop and finally a plateau in the ionic current as shown in Fig. 2a for three separate blocked and unblocked events. The two discrete ionic conductance levels, denoted by the red dashed lines in Fig. 2a, can be attributed to the unblocked and blocked states of the nanopore due to the presence of MNPs. The difference ΔI in ionic current between the unblocked and blocked states was ~500 pA for most of the blocking and unblocking events measured. And the typical $\tau_{block}$ and $\tau_{unblock}$ were found to be 2.7 ± 0.2 s and 0.04 ± 0.01 s (errors represent s.d., $n = 18$ for both), respectively (Fig. 2b). The ionic current can be repeatedly transitioned between the unblocked and blocked states for at least five cycles by changing the applied

magnetic field direction. This suggests the magnetically-controlled blocking and unblocking processes were highly reversible and reproducible.

**Pathways to blocking a nanopore**. We computed the electric potential and electric field distribution inside and local to a nanopore using COMSOL Multiphysics software (Fig. 3). The simulations feature a 27 nm nanopore with a truncated cone geometry in a 180 nm-thick SiN membrane. The SiN membrane was defined as an insulating region where no ionic current can pass through. The membrane and nanopore surfaces were modelled as coated with a 3 nm-thick organic self-assembled monolayer with a relative permittivity $\varepsilon_r = 2.5$. A potential bias V was applied across the SiN membrane with the top and bottom boundaries of the simulated region specified with potentials of +100 mV and 0 mV, respectively (Supplementary Figs. 12 and 13). The modelling predicts the ionic current remains constant (unblocked state) until the MNP comes within 250 nm of the nanopore, whereupon a gradual decrease in ionic current occurs (Fig. 3a(i–iii)). Thereafter, the ionic current drops drastically once the MNP enters the nanopore (Fig. 3a(iv)). The current becomes constant again once the MNP reaches the pore constriction and this plateau is defined as the blocked state (Fig. 3a(v)). The MNP remains trapped inside the pore as electrostatic forces from the applied potential bias push the positively-charged MNP against the constriction. This agrees well with the experimental results in Fig. 2a except that in the experiments the timescale for ionic current change as the MNP enters the nanopore is 70 times longer than the timescale over which the current changes as the MNP is ejected from the pore.

The difference in entry and exit times in Fig. 2a can be attributed to the difference in pathways for MNPs to enter and

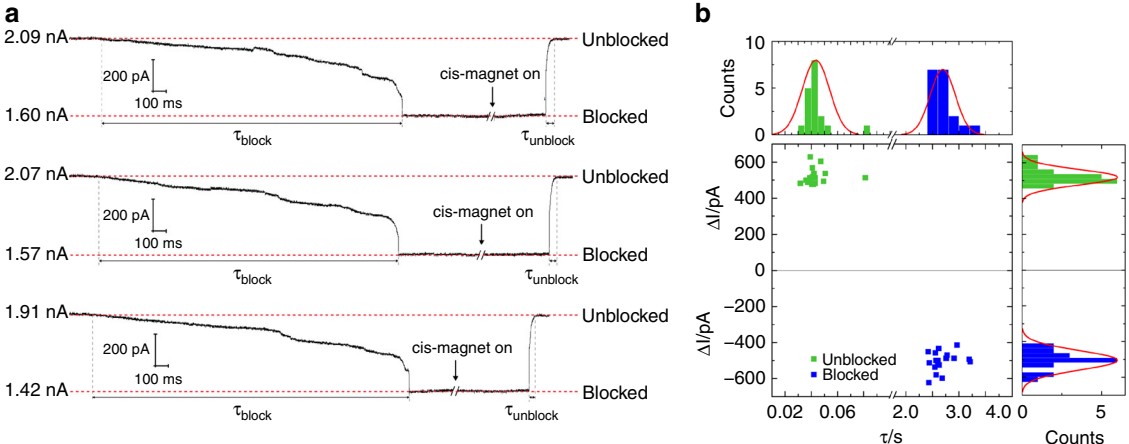

**Fig. 2** Controlled blocking and unblocking of nanopore with MNP. **a** Concatenated ionic current traces showing blocking and unblocking events of a 27 nm solid-state nanopore modified with silane-EG$_6$ SAM by 50 nm MNPs. 180-nm-thick SiN membranes were employed here. All traces were recorded in 100 mM KCl and 10 mM Tris pH 7.4 at a potential bias of 100 mV, low-pass-filtered at 10 kHz and sampled at 250 kHz. **b** Nanopore blocking and unblocking event statistics. Each data point in the scatter plot corresponds to the parameters of individual blocking and unblocking events ($n = 18$, respectively) measured from ionic current traces. ΔI and τ histograms are fitted with Gaussian distribution functions (red solid lines) with a bin size of 0.2 nA for ΔI and bin sizes of 0.2 s and 0.005 s for $τ_{block}$ and $τ_{unblock}$ respectively

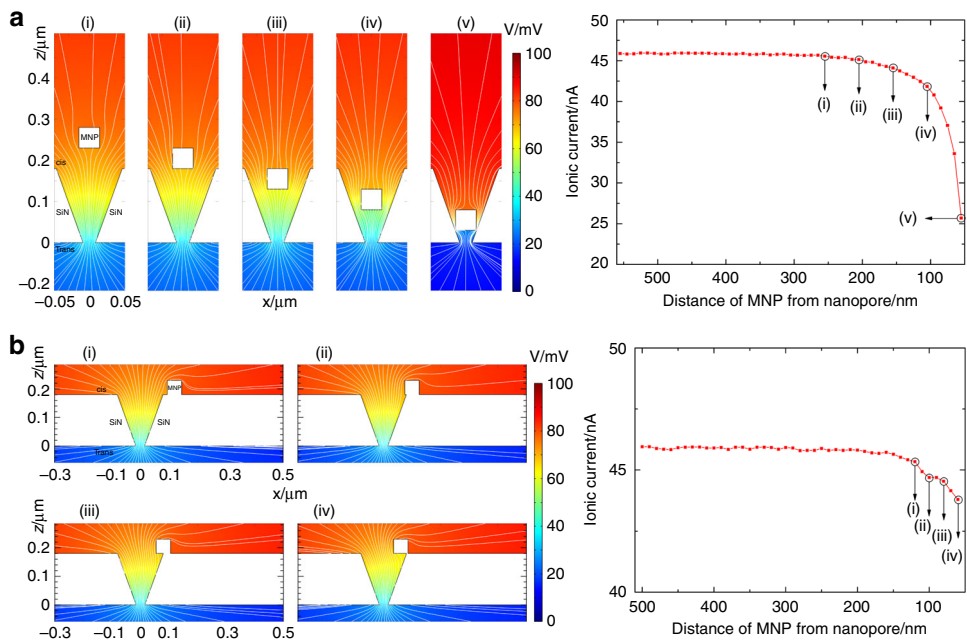

**Fig. 3** Pathways for a 50 nm MNP entering into a 30 nm nanopore. Electric potential and field distribution, as well as the corresponding approximated ionic currents as a function of distance related to the nanopore on a 180-nm-thick SiN membrane are present. **a** Direct entry. MNP is located at a distance of (i) 255 nm, (ii) 205 nm, (iii) 155 nm, (iv) 105 nm, (v) 55 nm away from the trans-side nanopore orifice. White solid lines represent the electric field lines in the electrolyte region. Approximated ionic current as a function of distance of the MNP from the trans-side nanopore orifice. **b** Indirect entry. The MNP is located at a distance of (i) 120 nm, (ii) 100 nm, (iii) 80 nm, (iv) 60 nm away from centre axis of the cis-side nanopore orifice. Electric potential distribution is colour coded in Volt (0~100 mV). White solid lines represent the electric field lines in the electrolyte region. Approximated ionic current as a function of distance of MNP from the centre axis of the cis-side nanopore orifice

exit the pore. Two main forces act on the MNPs: the magnetic force from the cis-/trans-magnet and the electrophoretic force. The magnetic force acts globally with uniform strength along the $z$-direction (positive $z$ for cis- and negative $z$ for trans-). In contrast, the electrophoretic force decays with increasing distance from the pore and acts somewhat radially, i.e., it can have significant $x$- and $y$-components. The two forces combine to give multiple pathways to enter the pore, but a single pathway for exit. A small number of MNPs will take the direct entry path in Fig. 3a. The more common alternative is the indirect entry path shown in

Fig. 3b. Here the magnetic force drives the MNP to the membrane adjacent to the pore, where it is then attracted towards the pore along the $x$-$y$ plane by the electrophoretic force. The COMSOL model predicts a rapid decrease in ionic current for 'direct entry' (Fig. 3a), since the MNP is rapidly driven directly into the pore by the combined magnetic and electrophoretic forces acting together along the path. The MNP will bind close to the pore constriction producing a higher ΔI. The indirect path gives a smaller ΔI and a slower decrease in ionic current. The former because the MNP will more likely bind to an anti-PSA

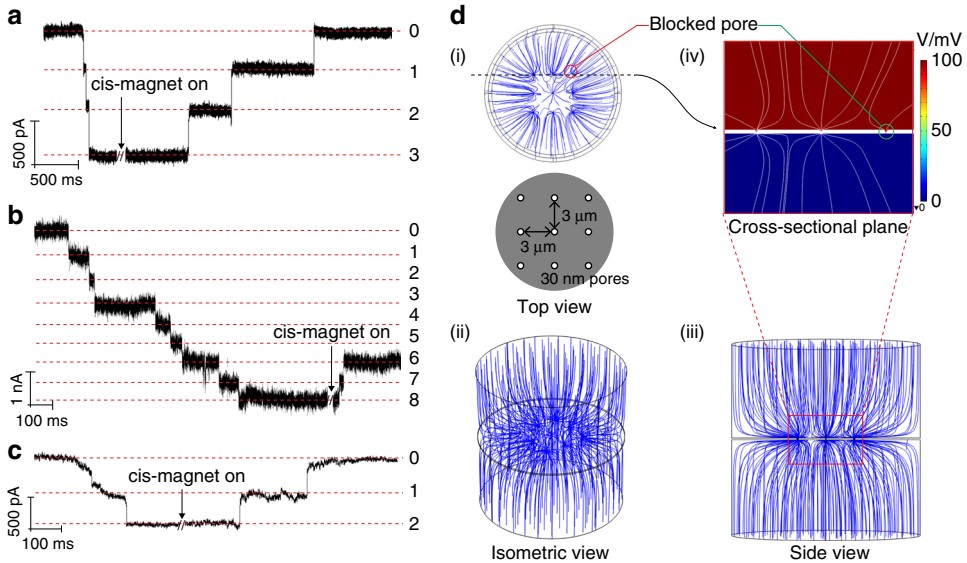

**Fig. 4** Paralleled detection of PSA using 3 × 3 nanopore array. Concatenated ionic current traces showing multiple sequential blocking and unblocking events of (**a**) 3 × 3 array of 27 nm nanopores modified with silane-EG$_6$ SAM by 50 nm MNPs when there was no PSA present. **b** 3 × 3 array of 27 nm nanopores modified with silane-EG$_6$ SAM by anti-PSA functionalised 50 nm MNPs in the presence of 5 pg mL$^{-1}$ PSA. **c** 3 × 3 array of 27 nm nanopores modified with silane-EG$_6$ SAM by anti-PSA functionalised 50 nm MNPs in the presence of 10 pg mL$^{-1}$ bovine serum albumin (BSA). All ionic current traces were recorded in 100 mM KCl and 10 mM Tris pH 7.4 at a potential bias of 100 mV, low-pass-filtered at 10 kHz and sampled at 250 kHz. (**d**) Simulation of the 3 × 3 array of 30 nm nanopores separated by 3 μm apart from each other. The simulation region consists of a cylindrical volume of 20 μm in diameter and 20.18 μm in height representing the electrolyte medium. A potential bias of +100 mV was applied from the top boundary whereas the bottom boundary was defined as grounded (0 mV). (i) Top, (ii) isometric, (iii) side view of 3D plots of electric field lines with one nanopore blocked by a 50 nm MNP. Black solid lines represent the interface boundaries whereas blue solid lines denote the simulated electric field lines. (iv) 2D cross-sectional plot of electrical potential and field lines of 3 nanopores (two unblocked and one blocked) as a colour map and white solid lines respectively. SiN membranes of a thickness of 180 nm were employed here for the paralleled detection and simulation work

higher in the nanopore. The latter because only the electrophoretic force acts to draw the MNP towards the nanopore; the magnetic force now only pushes the MNP against the membrane surface impeding drift. An indicative simulation for the pore blocking signal for the indirect path is shown in Fig. 3b, demonstrating the effect described above. Since the exit is driven by a reversal of the magnetic field direction, it is only possible via the reverse of the direct entry pathway. Thus, it will always have the same speed, set by the difference between the oppositely acting magnetic and electrophoretic forces. This explains the form of the experimental current data presented in Fig. 2a.

**Parallel detection using 3 by 3 nanopore array.** To demonstrate the potential of the nanopore blockade sensor for low detection limit analysis a 3 × 3 array of 27 nm nanopores was produced. Sequential blockade events occurred once the trans-magnet was switched on as shown in Fig. 4a. Switching on the cis-magnet results in sequential unblocking as MNPs are removed. In this case there was no PSA in solution and all blockade events are simply the MNPs entering the pores without specific binding. Here the nanopores in the array have been placed far enough apart (≫0.5 μm) such that the electric fields from adjacent nanopores will not interfere with each other. Thus the nanopores work independently and contribute to the change in ionic current a similar amount[31,32]. Eight blockade events were observed upon activating the trans-magnet for nanopores modified with silane-EG$_6$-(anti-PSA) when (anti-PSA)-MNPs with 5 pg mL$^{-1}$ (150 fM) PSA was added in the cis-side Fig. 4b. Two of the blocking MNPs were removed when the cis-magnet was switched on. These two removal events suggest there were six specific capture events and two non-specific capture events. To confirm the nanopore blockade sensor can differentiate non-specific

capture events from specific capture events, bovine serum albumin (BSA) was added to the cis electrolyte rather than PSA such that only non-specific blockade events were possible. In this case the two nanopores blocked by (anti-PSA)-MNP when the trans-magnet was applied were recovered upon switching to the cis-magnet (Fig. 4c). The removal of the (anti-PSA)-MNPs from the nanopores when no analyte is present also demonstrates the effectiveness of the hexa(ethylene oxide) surface chemistry from preventing non-specific adsorption of proteins to the SiN surface. Finally, no transients connected to translocation of proteins through the nanopores were detected in any of the measured ionic current traces. This is to some extent expected—both PSA and BSA have a net negative surface charge at pH 7.4[33,34] so the same electrophoretic force that attracts the positively charge MNPs should drive PSA and BSA away from the nanopores preventing their translocation.

Figure 4d represents a 3D simulation model for a SiN membrane with a 3 × 3 array of 30 nm nanopores with 3 μm spacing where the top-right nanopore is blocked. The solid blue lines in Fig. 4d(i–iii) represent the electric field lines local to the nanopores. Figure 4d(i) demonstrates that the pores located at the array edge will have a higher probability of being blocked due to the higher number of field lines going through these pores. When a nanopore is blocked, the number of electric field lines going through the pore decreases relative to adjacent pores (Fig. 4d(iv)). This agrees with the reduced ionic current measured experimentally in Fig. 4(a). Correspondingly, if one pore in an array is blocked, the subsequent incoming MNPs will be preferentially attracted towards the closest adjacent open pore instead. It is thus possible to use arrays of nanopores in sensing to detect multiple blocking and unblocking events simultaneously, which will facilitate in using nanopore blockade sensors for quantitative analysis.

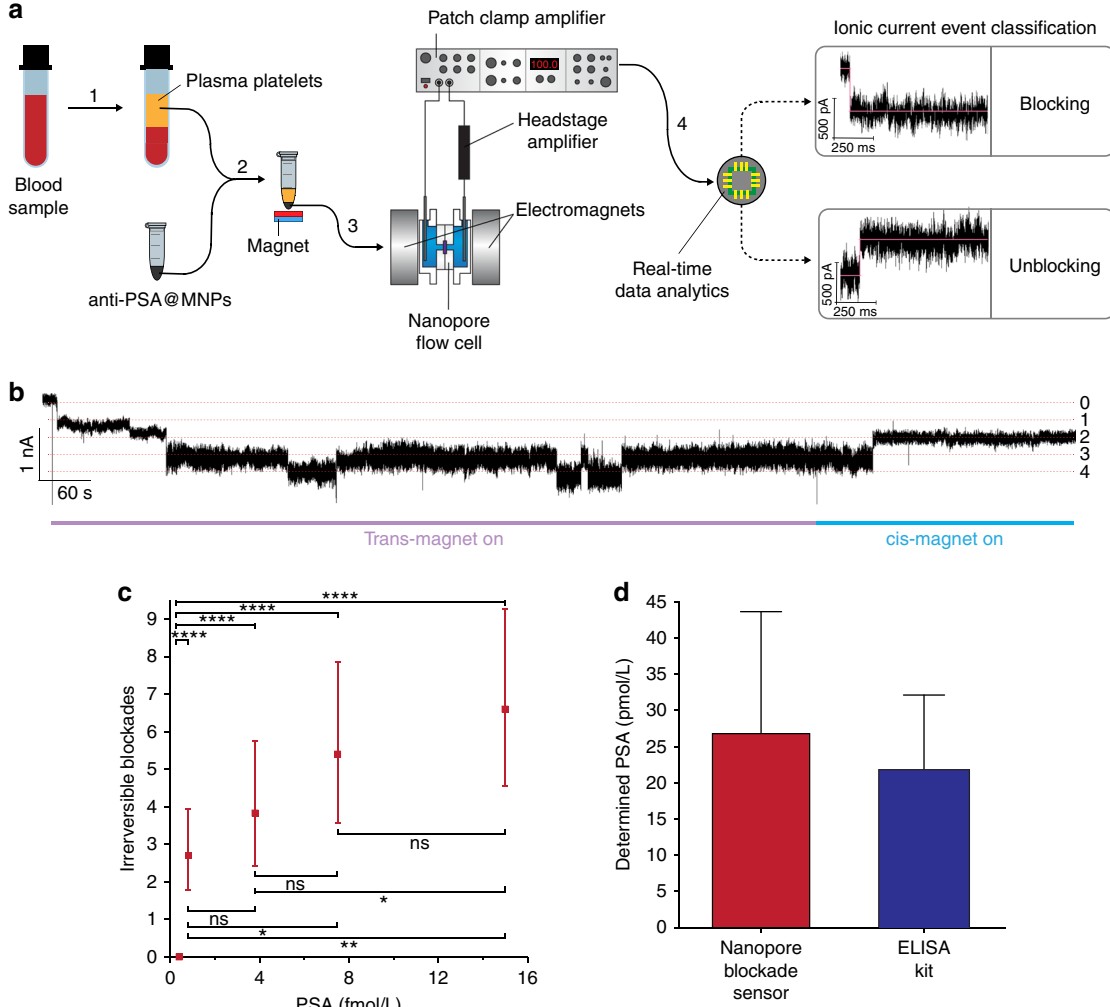

**Fig. 5** PSA detection from blood sample. **a** Flow chart illustrates steps for detecting PSA in blood sample: (1) Plasma platelets were separated from blood with centrifugation. (2) The plasma platelet layer was extracted and mixed with (anti-PSA)-MNPs. The (anti-PSA)-MNPs were magnetically separated and washed with PBS. (3) The (anti-PSA)-MNPs were added into cis-chamber of a flow cell. A potential difference was applied between both sides of the nanopore and a magnetic field was applied from trans-side of flow cell to initialise detection process. (4) Ionic current measured by patch clamp amplifier was analysed by an event classifier where three distinct event types—translocation, blocking and unblocking are detected and classified as they occur. This enables tracking of the number of blocking and unblocking events that occurred over the course of analysis. For application here in quantitative analysis, SiN membranes have been thinned down to about 80 nm. **b** Ionic current trace from one experiment that employed (anti-PSA)-MNPs to capture PSA from whole blood and transferred to the $3 \times 3$ array nanopore blockade sensor. The trace shows four blockade events but upon switching on the cis-magnet it transpires two of these blockade events are non-specific. **c** Irreversible blockades observed versus PSA concentrations. Blockade events were counted after 4 cycles of switching on trans-magnet for 10 min, and subsequently cis-magnet for 5 min to maximise probability of blockade events at extremely low concentrations. Mean value of irreversible blockades and 95% exact confidence intervals were obtained from a total of 31 measurements ($n = 5, 10, 6, 5$ and 5 from lower to higher concentrations; one measurement per chip). Of these measurements, 23 chips were employed and 8 of them were regenerated. The exact method was used to compute 95% confidence intervals for Poisson means. The difference between any two determined Poisson means was tested by a two-tailed z-test of square root transform version; ns = not significant, $*p \leq 0.05$, $**p \leq 0.01$, $***p \leq 0.001$ and $****p \leq 0.0001$. **d** Comparison of PSA level from whole blood using nanopore blockade sensor versus an ELISA kit. Error bars represent uncertainty of determined PSA concentration at 95% confidence by nanopore sensing ($n = 6$) and ELISA kit ($n = 12$), respectively

Finally, the analytical performance of the $3 \times 3$ array was assessed. The ability of the sensor to monitor low concentrations of PSA is shown in Fig. 5a. The cis- and trans-magnets were toggled on and off for four cycles to maximise the probability of (anti-PSA)-MNP capture events; each cycle lasted 15 min. Most importantly, the lowest detected concentration was 0.8 fM could be detected even with only nine nanopores. At lower concentrations no blockade events were observed. That is an order of magnitude lower than the current best performance of ~5 fM reported by Freedman et al.[8] Even lower detectable concentrations could be achieved with more pores or more magnetic switching cycles. At 0.8 fM PSA concentration there are $3.65 \times 10^5$ PSA molecules in the reaction cell and $1.87 \times 10^9$ (anti-PSA)-MNPs. By Poisson statistics all but 36 of the PSA molecules will end up singly-captured by a MNP, which will be 1.95% of the total population of MNPs; there will be only 18 MNPs with two PSA molecules bound. Hence 99.99995% of blockade events will be attributed to single-molecule analyte capture, indicating high quantitative accuracy.

At higher concentrations of PSA the calibration plot shows more blockade events as the concentration increases. It should be noted however, that at these low concentrations there is some

variability in the number of blockade events for each concentration. With only nine nanopores and the digital response obtained with this system, the uncertainties are such that the calibration plot is only semi-quantitative; although it is expected with greater number of pores lower uncertainties would be achieved. It is noted here that with 10 measurements at 0.8 fM the 95% confidence intervals suggest that increasing the number of repeated measurements helps to reduce uncertainties as well. An additional important point here regards the selectivity of the system that is provided by the use of two antibodies to detect the analyte. As has been demonstrated previously the sandwich format gives very high selectivity[35]. It is possible however that an antigen might evade capture by the (anti-PSA)-MNP but bind to antibodies within the nanopore. However, with the large size of the nanopores, such an event would cause only a small drop in current relative to the current decrease during a blockade event. Similarly, small variations in nanopore diameter will give slight differences in current but again the changes observed with blockade events will be much larger. Hence small variations in MNP size or nanopore size will have minimal impact of the robustness of the measurement.

**Sensing protein analyte in whole blood**. Analysis of PSA in whole blood was performed to further demonstrate the analytical utility of our nanopore blockade sensor (Fig. 5). The analysis was performed by first incubating the (anti-PSA)-MNP nanoparticles in the blood, followed by separating and washing the (anti-PSA)-MNPs and transferring them to the cis side of the nanopore reaction cell. An example of the ionic current blockade is shown in Fig. 5b where four blockade events were observed. Switching on the cis-magnet to pull the (anti-PSA)-MNPs away from the nanopore shows that two of these were (anti-PSA)-MNPs that had captured PSA, and hence were not removed. A correlation between the number of detected irreversible blockades and PSA levels had been performed (Fig. 5c). A monotonic trend was found that when a higher concentration of PSA was present, the mean number of detected irreversible blockades increased as well, albeit with significant uncertainties with only nine nanopores. Interestingly, there was no irreversible blockade detected when there was 0.4 fM PSA present in the electrolyte solution. Afterwards, the nanopore blockade sensors were challenged with plasma samples that had been diluted to have even lower concentration of PSA. Importantly, the concentration of PSA determined by the nanopore blockade sensor correlated well with that obtained by a commercial enzyme-linked immunosorbent assay (ELISA) kit for the same whole blood sample (Fig. 5d).

## Discussion

In this study, we demonstrated a nanopore sensor capable of detecting single-molecule-level events at an ultralow detection limit of 0.8 fM in complex biological media. The device features antibody-labelled magnetic nanoparticles, which capture analyte, and are driven by an external magnetic field towards an antibody-labelled nanopore (different epitope) where they form pore-blocking sandwich complexes to provide specificity, sub-fM detection limits with rapid response times. The blockades give robust analytical signals where large changes in current are observed as each nanopore is blocked such that small variations in pore size during fabrication, or proteins translocating through the nanopores without an associated magnetic nanoparticle do not give false signals.

This represents a radical change in nanopore sensor paradigm, going from passive sensing where one must wait for the analyte to find the pore to an active sensing mechanism where analytes are 'magnetically shuttled' to the sensor point. This paradigm shift is important as it allows significantly lower analyte concentrations to be detected with more rapid throughput. Our detection limit of 0.8 fM surpasses previous reports for nanopore sensors by almost an order of magnitude. Another important attribute of our design is that active magnetic filtering steps via repeated magnetic field-reversals can be introduced to eliminate non-specific blockades. That is, nanoparticles that block the pores non-specifically are removed by reversing the magnetic field while if the immuno-sandwich is formed the magnetic nanoparticles cannot be removed. Thereby detection becomes highly specific, enabling the use of nanopore sensors for monitoring protein analytes from whole blood for the first time. Performing the measurement in complex biological media does mean that other proteins within the sample could translocate through the pore but will not block it as the nanopore is too large, ~30 nm at its smallest orifice. As such, these translocation events may increase the background noise but will not be counted as a binding event as any change in current is not permanent as it is when the immuno-sandwich is formed. The changes in the nanopore sensing paradigm therefore really opens the door towards the commercialisation of nanopores for quantitative analysis where devices have ultralow detection limits and are compatible with both bulk manufacture and the fabrication of portable devices.

One potential concern for nanopore blockade sensors is if other species bind to the antibodies within the nanopores. It seems plausible that such binding events may decrease the ionic current. However, with the nanopores being 30 nm in diameter at its minimum, a single protein binding event is unlikely to cause a current drop of the magnitude associated with the nanoparticle binding to the nanopore. In the nanopore blockade sensor format presented herein, the capture of the analyte is by the (anti-PSA)-MNPs and the nanopore measurement are spatially separated.

## Methods

**Chemicals and materials**. All chemicals were of analytical grade and used as received. Four inch double-side polished 105 μm-thick undoped silicon (<100> ± 0.9°) wafers (>20 Ω cm resistivity) deposited with 180-nm-thick SiN layer on both sides were purchased from Virginia Semiconductor Inc. (USA). Potassium chloride (KCl), N,N′-disuccinimidyl carbonate (DSC) and 4-dimethylaminopyridine (DMAP), TWEEN20, ethylene glycol-bis(succinic acid N-hydroxysuccimide ester) (EGS), tris(hydroxymethyl)aminomethane (tris base), Dulbecco's phosphate-buffered saline (PBS 10×, without calcium chloride and magnesium chloride), sodium hydroxide (NaOH), potassium nitrate (KNO₃), iron (II) sulfate heptahydrate, polyethylene(imine) (branched, 25 kDa) and bovine serum albumin (BSA) were obtained from Sigma-Aldrich. ZEP520A electron beam resist was purchased from ZEON Chemicals (Japan). Custom synthesised silane with a molecular formula $(CH_3O)_3-Si-(CH_2)_3-EG_6-OSi(CH_3)_3$ where EG is abbreviated for ethylene glycol, was purchased from ProChimia Surfaces (Poland). Human PSA and anti-PSA antibodies (catalogue numbers ab10187 and ab10185) were purchased from AbCam (UK). The received stock solutions for these two antibodies (catalogue numbers: ab10185 and ab10187) were diluted 1:10 (v/v) and 1:200 (v/v) in phosphate-buffered saline (PBS 1×, pH 7.4) solution immediately prior to use. Acetonitrile, dimethylformamide (DMF), potassium hydroxide (KOH), methanol were purchased from Ajax Finechem. Human whole blood (anticoagulated with K2 EDTA) was purchased from Innovative Research (Novi, USA). Free prostate-specific antigen (f-PSA) ELISA kit was obtained from GenWay Biotech Inc (USA). Polydimethylsiloxane (PDMS) films were made by using SYLGARD 184 silicone elastomer kit (Dow Corning, USA). N-methyl-2-pyrrolidone (NMP) and N-amyl acetate were purchased from Merck Millipore. Milli-Q water (~18 MΩ cm, Millipore, Australia) was used to prepare experimental solutions.

**SiN nanopore array fabrication and characterisation**. A 105 μm-thick silicon <100> wafer was deposited with 180 nm-thick SiN layer on both sides via low-pressure chemical vapour deposition (LPCVD). The wafer was subsequently patterned via photolithography and etched using $CF_4$ RIE followed by wet chemical etching in 45% w/w KOH solution at 85 °C to create 180 nm thin freestanding SiN membranes. The SiN chip was then spin-coated with ZEP520A onto the frontside and exposed with an EBL system (Raith 150-TWO) using dot exposure mode to pattern the nanopore array. The pattern was developed with N-amyl acetate and etched using $CF_4$ plasma followed by removal of resist using NMP.

**Chemical modification of SiN nanopore array**. A solid-state nanopore array chip was cleaned with oxygen plasma for 5 min followed by rinsing with deionized water and filtered methanol and drying at 100 °C to remove organic contaminants and increase the amount of oxide species on the surface. The chip was then immersed into a 2 mM custom synthesised silane with molecular formula $(CH_3O)_3$-Si-$(CH_2)_3$-$EG_6$-$OSi(CH_3)_3$ in dry toluene solution for 2 h, followed by rinsing with methanol for 10 min to remove the -$OSi(CH_3)_3$ protective group, dried under $N_2$ and baked at 120 °C for 30 min.

**Coupling of anti-PSA antibodies onto nanopore array**. A nanopore chip containing an array of silane-$EG_6$ modified solid-state nanopore was immersed into 100 mM N,N′-disuccinimidyl carbonate (DSC) and 100 mM 4-dimethylaminopyridine (DMAP) dissolved in dry acetonitrile solution for 8 h and rinsed with dry acetonitrile. The chip was then immersed in 10 µg mL$^{-1}$ capture anti-PSA in phosphate-buffered saline (PBS 1×, pH 7.4) solution for 90 min and rinsed with PBS. Anti-PSA mouse monoclonal antibodies were from AbCam (catalogue number ab10187) and were specific for free PSA, epitope 1.

**Iron oxide magnetic nanoparticles**. The synthesis and characterisation of iron oxide magnetic nanoparticles follow a previous study[36]. Briefly, 2 M of $KNO_3$ and 1 M of NaOH were added into $FeSO_4$ in an oxygen-free environment to produce a green precipitate. The suspension was heated to 90 °C for 2 h and the dark brown particles were collected using a neodymium magnet and washed three times with deionized water. Next, the magnetic nanoparticles were added into a 20 mL of 20 g L$^{-1}$ polyethyleneimine (PEI) solution followed by sonication. The mixture was left at 90 °C for 4 h and the PEI-coated magnetic nanoparticles were washed three times and resuspended in deionized water to its original volume. The zeta potential, $\zeta$ and electrophoretic mobility, $\mu_e$ of the MNPs were found to be +12 mV and 0.9 µm cm V$^{-1}$ s$^{-1}$ respectively in 100 mM KCl at pH 7.4.

A two-step crosslinking method was used to immobilise anti-PSA mouse monoclonal antibody (catalogue number ab10185, AbCam), which were specific for PSA epitope 5, onto the surface of the PEI-coated MNPs. This involves coupling one end of the ethylene glycol-bis(succinic acid N-hydroxysuccinimide ester) (EGS) crosslinker to an anti-PSA antibody and followed conjugation of the other end of the EGS-(anti-PSA) complex to the amine-rich PEI-coated MNP via formation of an amide bond. Briefly, 10 µL of 24 µM EGS in DMF was mixed with 90 µL of anti-PSA antibody (200 µg mL$^{-1}$) in phosphate-buffered saline (PBS 1×, pH 7.4) solution. After 30 min, 25 µL of PEI-coated MNPs suspension (2.4 mg mL$^{-1}$) was introduced into the above solution. The mixture was kept in dark under room temperature for 2 h. Nanoparticles were collected by using a magnet, rinsed with PBS, resuspended in 1 mL of PBS and stored in fridge at 4 °C.

**Antibody specificity**. The anti-PSA antibodies used in this study were the mouse IgG monoclonal anti-PSA specific antibodies, with one anti-PSA antibody (ab10187) specific for epitope 1 (only free PSA) while the other anti-PSA antibody (ab10185) is specific for epitope 5 of PSA (both free PSA and complexed PSA)[37,38].

**Finite-element analysis**. COMSOL Multiphysics (version 4.4) was used in modelling the ionic current density inside the nanopores as a function of the distance between a magnetic nanoparticle and a nanopore. Three software modules were used: electrostatics (AC/DC Module), transport of diluted species (Chemical Species Transport Module) for the calculations of K$^+$ ions and Cl$^-$ ions, and laminar flow (Fluidic Flow Module). The resulting ionic current was computed by integrating the flux density along the boundary of the reservoir. The following physical parameters were used in the simulation model: relative permittivity $\varepsilon_r = 77$, K$^+$ ion mobility $\mu_K = 7.8 \times 10^{-8}$ m$^2$ V s$^{-1}$, Cl$^-$ ion mobility $\mu_{Cl} = 7.909 \times 10^{-8}$ m$^2$ V s$^{-1}$, diffusion constant of K$^+$ ions, $D_K = 1.957 \times 10^{-9}$ m$^2$ s$^{-1}$, diffusion constant of Cl$^-$ ions $D_{Cl} = 2.032 \times 10^{-9}$ m$^2$ s$^{-1}$, fluidic density $\rho = 1000$ kg m$^{-3}$, dynamic viscosity $\mu = 8.91 \times 10^{-4}$ Pa S, and the surface charge density of SiN wall at pH 8.0 $\rho_{wall} = -49$ mC m$^{-2}$ [39]. The Debye length was estimated to be 0.95 nm, $\varepsilon_r = 77$ for 0.1 M KCl. The nanopore was modelled to have a truncated cone shape in all simulations to match the actual nanopore geometry profile measured by TEM tomography. Details of the finite-element simulation model are provided in the Supplementary Information.

**Nanopore data acquisition and analysis**. A chemically-modified solid-state nanopore array chip was mounted between two compartments of a custom-made flow cell filled with degassed and filtered (0.2 µm, Millipore) 100 mM KCl and 10 mM tris(hydroxymethyl)aminomethane (Tris)-base pH 7.4 solution. Two 1.5-inch round electromagnets (APW Company, USA) mounted on each adjacent side of the flow cell were connected to a programmable power supply unit (Hioki) and switched using a custom-written computer program. Two Ag/AgCl electrodes were inserted into the flow cell and connected to an Axopatch 200B patch clamp amplifier (Axon Instruments, Molecular Devices) in voltage clamp mode. The ionic current traces were filtered with a built-in 4-pole low-pass Bessel filter (80 dB/decade) at 10 kHz and acquired with Clampex (version 10.4, Axon Instruments) via a DigiData 1440A (Axon Instruments, Molecular Devices) analog-to-digital

converter at 250 kHz sampling rate. Clampfit (version 10.4, Axon Instruments, Molecular Devices) was used to analyse the ionic current traces.

The chips were sandwiched by two PDMS gaskets and the flow cell was assembled using four hex nuts and bolts forming a tight seal around the chips such that the nanopores are the only liquid connector for the cis- and trans-side of the flow cell compartments. The 100 mM KCl (with 0.05% v/v Tween-20 surfactant and 10 mM Tris, pH 7.4) solutions were prepared using deionized water, degassed under low vacuum condition for at least 1 h and filtered with syringe filter (0.22 µm, Merck Millipore Ltd). Samples of a 20-µL solution volume were prepared by mixing 4 µL of (anti-PSA)-MNPs with 16 µL of synthetic solutions of PSA of various concentrations or human whole blood samples. The mixed samples were kept at room temperature on a WiseMix®RT programmable digital rotator rotating at 25 revolutions per minute for about 60 min. Afterwards the magnetic particles were collected by applying a magnet, washed three times and dispersed in 20 µL of KCl electrolyte solution. This obtained 20 µL sample together with 780 µL of KCl electrolyte were loaded into the cis-side chamber of the assembled flow cell while the trans-side chamber was filled with 800 µL of KCl electrolyte solution.

A potential difference of 100 mV was applied between the two Ag/AgCl electrodes. Current-time traces were recorded using an Axopatch 200B patch clamp amplifier and a 1440 A Digidata (Molecular Devices, Inc.) at 250 kHz sampling frequency and 10 kHz low-pass Bessel filter while switching on the trans-side magnet for 10 min and afterwards the cis-side magnet was switched on whereas the trans-side magnet was turned off. Multiple switching on/off for the cis-side and trans-side magnets were performed. Information about events like number of blocking and unblocking steps and specific blockades were analysed and extracted from the ionic current traces. After ionic current experiment, these used chips can be regenerated for multiple usages. The nanoparticle induced pore blockades can be unblocked by applying an external strong magnet to draw the MNPs out of the nanopore. After dark-field optical microscopy is used to confirm that nanopores are in opened state, the chips were cleaned with 10 min in an oxygen plasma to both sides of the chip. The chip can then be re-modified using the same procedure as described in the chip modification section of the paper. Prior to being used, a careful examination on the measured resistance and noise level was performed to ensure these regenerated chips behaviour comparably to the non-regenerated (fresh) chips.

**Reporting summary**. Further information on research design is available in the Nature Research Reporting Summary linked to this article.

## Data availability

The authors declare that all the data supporting the findings of this study are available within this paper and its Supplementary Information file, or available from the corresponding author upon reasonable request.

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

## Acknowledgements

We acknowledge the generous financial support from the Australian Research Council Centre of Excellence in Convergent Bio-Nano Science and Technology (CE140100036, J.J.G.), a Discovery Project (DP170104024, A.P.M.), an ARC Australian Laureate Fellowship (FL150100060, J.J.G.) and an ARC Future Fellowship (FT0990285, A.P.M.). Yanfang Wu appreciates financial support by an Australian Government Research Training Program Scholarship. This work was performed in part at the NSW Node of the Australian National Fabrication Facility (ANFF). The authors are grateful to Dr. Muhammand Tanzirul Alam and Dr. Alexander H. Soeriyadi for helpful discussions.

## Author contributions

K.C., Y.W. and J.J.G. designed the experiments, performed data interpretation and wrote the manuscript. K.C. and Y.W. performed the experiments. A.P.M., P.J.R. and K.G. assisted with building the experiment and interpreting the data, S.R.C.V. assisted with the computational modelling. All authors reviewed the manuscript.
