## [Peer Review File · Nature Communications]

Reviewer #1 (Remarks to the Author):

It is an interesting work by combining the nanopore sensor and the magnetic manipulation of the MNPs to enter nanopores for detecting PSA. Its detection limit can reach sub-fM based on their estimation.

Major revision is required for a final decision is reached. Please address the following comments.

1. How to verify the surface of the sidewall of SiN nanopores can be chemically modified in Fig. 1a(ii)?
2. Obviously the surface of SiN membrane surrounding the nanopores also has been chemically modified, does this affect the measured ion current? Especially a lot of more MNPs would be bound to the membrane than the sidewall of the nanopore.
3. Improve the quality of Fig. 2. The font size of the text is too small, etc.
4. Line 176, the blocked state should be in Fig. 3(v)? Please clarify.
5. How many MNPs could be bound to the sidewall of the nanopore, given the thickness of the SiN is 180 nm (Fig. S1)? Does the number of MNPs bound to the surface of the nanopore affect the ion current?
6. If possible, please have SEM images or TEM images to show the MNPs are in the nanopores.
7. In line 269-270, "It is thus possible to detect large numbers of blocking and unblocking events simultaneously because most of the pores likely capture only one nanoparticle at a time". Any evidence or reference?
8. Please add more references of micro-nanodevices for detecting PSA. Compare the **device fabrication** advantages and disadvantages with them. As we know EBL is quite expensive to fabricate nanopores. In addition, is it necessary to detect PSA at sub-fM concentration for real medical practice?

- (1) Damborska, D., Bertok, T., Dosekova, E., Holazova, A., Lorencova, L., Kasak, P., & Tkac, J. (2017). Nanomaterial-based biosensors for detection of prostate specific antigen. *Microchimica Acta*, 184(9), 3049-3067.
- (2) Stern, E., Vacic, A., Rajan, N. K., Criscione, J. M., Park, J., Ilic, B. R., ... & Fahmy, T. M. (2010). Label-free biomarker detection from whole blood. *Nature nanotechnology*, 5(2), 138.
- (3) Alzghoul, S., Hailat, M., Zivanovic, S., Que, L., & Shah, G. V. (2016). Measurement of serum prostate cancer markers using a nanopore thin film based optofluidic chip. *Biosensors and Bioelectronics*, 77, 491-498.
- (4) Kosaka, P. M., Pini, V., Ruz, J. J., Da Silva, R. A., González, M. U., Ramos, D., ... & Tamayo, J. (2014). Detection of cancer biomarkers in serum using a hybrid mechanical and optoplasmonic nanosensor. *Nature nanotechnology*, 9(12), 1047.
- (5) Zheng, G., Patolsky, F., Cui, Y., Wang, W. U., & Lieber, C. M. (2005). Multiplexed electrical detection of cancer markers with nanowire sensor arrays. *Nature biotechnology*, 23(10), 1294.

9. The authors use the number of blockage events to assess the detection limit of the nanopore sensors. This is very interesting. Why amplitude of ionic current change is not used to estimate the detection limit? Based on authors' statement, ideally one nanopore can capture one MNPs, leading to one blockage event. Is it possible that one nanopore can capture two or more MNPs? If so, how about the accuracy to estimate the detection limit?

10. More explanations of Figure 5.c are needed in the text.

Reviewer #2 (Remarks to the Author):

Overall, the authors begin with a good review of the current status of the field in the use of nanopore sensors for analyte detection, the existing limitations, and some of the recent advances for improving the detection limit to the pM and the need to push further to the fM scale. The authors are presenting a technique to improve the sensitivity and specificity of using nanopores for biosensing in particular by using antibody tagged magnetic particle that will attach to the bioanalyte (in this case prostate specific antigen) as well as antibodies that have been functionalized to the inner surface of the nanopore itself. By subsequently applying an external magnetic field the PSA + antibody tagged MNP (epitope 5) is then drawn to the nanopores to accelerate the potential for the 2nd binding event (via epitope 1). The magnetic field is then applied in the opposite direction to remove any PSA + antibody tagged MNPs that did not end up binding to the nanopores with antibodies. This active process supersedes the previously reported passive process of waiting for the bioanalyte to travel and block the nanopore and also improves the specificity based on antibody binding bypassing previous concerns where larger biomolecules that were not the bioanalyte of interest would be able to pass and block the nanopores.

1. The authors should discuss the specificity of epitopes 1 and 5 chosen for targeting PSA and if there's any known nonspecific binding of the antibodies they used to target these specific epitopes. The authors should also discuss or quantify the number of antibody sites per nanopore and per MNP.
2. However, this currently reported method still would have the issue with larger biomolecules that passively pass by and block the nanopore. Although the authors do report that the electrophoretic charges would dispel the attraction of negatively charged molecules such as BSA and PSA and other proteins to the negatively charged nanopore surface. This was further validated when using a human blood sample where the plasma platelets were first isolated before incubation with the MNPs. Further discussion on how less negatively charged biomolecules or positively charged biomolecules could impact the technology. Some added discussion on what cases would the magnetic forces overcome the electrophoretic forces and vice versa and which would play the more critical role in the determination of the combined analyte + MNP being able to make it into the nanopore for complete blocking.
3. The size of nanopores were specifically customized to allow for the MNP to pass through but not translocate through the nanopore. Using a patch clamp setup, they were able to measure the number of nanopores that were now blocked to the quantify the number of PSA molecules detected. However, the authors also discussed the potential impact of how the MNP traveled into the nanopore could impact the ability for PSA+MNP to block the nanopore properly. It would be helpful for the authors to have added discussion about how to design and optimize the pattern of nanopores for the biological sample (i.e. how far apart or the minimum distance between the nanopores to still maintain the sensitivity of molecule detection – is there a distance small enough at which a PSA+MNP becomes stuck between two open nanopores and unable to make its way into one of them).
4. Similarly, although this appears to be a great tool for low levels of detection of PSA. The authors should discuss how these new concentration levels compare to the physiological levels of PSA, and what levels are important for diagnostic purposes such as in prostate cancer or benign prostatic hyperplasia (BPH). As a biomarker, PSA already provides a significant number of false positives and is primarily used as a trending biomarker and not a diagnostic biomarker (i.e. knowing that we have a very low level of PSA is not particularly meaningful and knowing the rate at which the PSA levels have changed is more diagnostically important).

This new report certainly provides a new leap forward in bionanosensing using nanopores by moving from passive detection to active detection. However, the technology's specificity becomes limited to the specificity of the antibody to the epitopes of the PSA. Nonspecific binding by these antibodies could potentially contribute to mis-capturing and counting. Some further discussion on this would be helpful. Overall, this is a well written paper with good mix between the theoretical /

computational aspects and the experimental aspects of nanopores, presents a new view in biosensing using nanopores, and provides an initial good proof of principle demonstration of the reported platform technology.

Reviewer #3 (Remarks to the Author):

Chuah et al. present data describing the use of immunomagnetic nanoparticles to enhance the sensitivity and specificity of analyte detection in a nanopore-based detection system. Their data indicates that magnetic fields can be used to directly immunomagnetic beads towards an antibody conjugated nanopore membrane where those that have captured the antigen of interest will be retained when the magnetic field polarity is switched, while nanoparticles that have not bound antigen will be removed from the membrane. Notably, the retained nanoparticles are interact with and block the current flow through the nanopore membrane in correspondence with the relative abundance of the target antigen. Improving the sensitivity of nanopore-based detection approaches is highly desirable; however, there are several shortcomings with the current study that greatly temper enthusiasm for its publication.

First, while the author presents data that they can detect an antigen target at low concentration, it is not clear that this system can accurately distinguish different concentrations of target from a given sample or how reliable these measurements would be in a panel of human samples. Specifically, Figure 5C indicates that there is considerable overlap between the values obtained at each concentration and there appears to be a very limited detection range. This data is not adequately discussed in the text or figure legends. Does this data represent the mean of the samples? If so, how many samples were analyzed at each point? Do the error bars indicate standard deviations or the standard error of the mean? Were there significant differences between the values at each concentration?

There is also not sufficient experimental to evaluate the utility of method. Nanoparticles are drawn to the membrane with a magnetic field that is cycled four times to allow specific enrichment of the magnetic nanoparticles that have bound the target protein, while the antigen-free nanoparticles are removed when the magnetic field is inverted to draw the particles away from the membrane. However, the authors do not describe whether the nanopore membrane can be regenerated by application of a stronger magnetic field or other method to allow reanalysis of the sample. Repeated measurements are needed to evaluate the accuracy and reproducibility of the assay results.

While I find this research intriguing, these fundamental questions need to be answered before the manuscript could be judged acceptable for publication.

Thank you very much for your comments concerning on our manuscript entitled **“Nanopore Blockade Sensors for Ultrasensitive Detection of Proteins in Complex Biological Samples”** (ID: NCOMMS-18-13751). These comments are valuable and helpful. With these comments, we have revised our manuscript carefully and those changes made in the revised manuscript and supplementary information have been highlighted with red colour. Our responses to these comments that the referees have raised are given below as attached.

Answers to referee #1:

Comment 1: How to verify the surface of the sidewall of SiN nanopores can be chemically modified in Fig. 1a(ii)? □

Authors' response: The referee raises an interesting point. We are currently not aware of methods that can unambiguously determine the chemical nature of the surface chemistry inside the pore alone. We have provided XPS data to show that the surface chemistry works on the entire surface in the Supplementary Figure 6. Given the well understood nature of organosilane chemistry, and that only one surface modifying species is employed, it is logical to assume the chemistry inside the pore is the same as on the exterior surface of the chip containing the nanopore. We did however do extensive characterisation of the electrical properties of the nanopores before and after surface modification to verify the surface chemistry was inside the nanopore. This was included in the original submission in Supplementary Figures 7 and 8 and is discussed at length below Supplementary Table 1. What the electrical data shows is a clear difference before and after modification with an increase in ionic resistance which is consistent with an organic layer reducing the nanopore diameter by 5.7 nm in length which is consistent with a molecule so of ~2.77 nm attached to the entire circumference of the nanopore.

To ensure this is clear, the following sentence has been added to page 5, lines 28 and page 6, lines 1-2

“The change in the ionic current suggests a reduction in the nanopore diameter of ~5.7 nm, consistent with the surface of the nanopore being modified with the silane-EG₆ species of length ~2.77 nm.”

Comment 2: Obviously the surface of SiN membrane surrounding the nanopores also has been chemically modified, does this affect the measured ion current? Especially a lot of more MNPs would be bound to the membrane than the sidewall of the nanopore.

Authors' response: It is correct that the rest of the topside surface outside of nanopores also have been modified with the analyte specific receptors – i.e., anti-PSA antibodies (specific to epitope 1 on PSA molecule). However, this does not affect the usage of the monitored current reductions and the steps for the quantification of PSA concentrations. One of the advantageous features about the nanopore blockade sensors here is that the diameter of the nanopores are relatively large (bottom orifice, ~30 nm; top orifice, ~88 nm; membrane thickness, ~80 nm) compared with other nanopore sensors. As such there are large changes in current (few hundreds of pA) from nanoparticle-induced pore blockades which means changes in surface chemistry on the exterior surface will not influence the measured ionic current changes as the measurement is based on the numbers of these stepwise drops in current, but not the magnitude of ionic current for these steps.

Comment 3: Improve the quality of Fig. 2. The font size of the text is too small, etc.

Authors' response: According to the referee's comment, the revision on Fig. 2 has been done.

□

Comment 4: Line 176, the blocked state should be in Fig. 3(v)? Please clarify. □

Authors' response: Thank you for the comment. We have corrected this error.

See page 8, line 7

“Fig. 3a(iv)” has been revised as “Fig. 3a(v)”.

Comment 5: How many MNPs could be bound to the sidewall of the nanopore, given the thickness of the SiN is 180 nm (Fig. S1)? Does the number of MNPs bound to the surface of the nanopore affect the ion current? □

Authors' response: For the sensing experiments, the SiN membranes have been thinned by reactive ion etching down to about 80 nm with typical geometrical dimensions for bottom and topside orifices of ~30 nm and ~88 nm, respectively. The nanopore was designed in this way to ensure only 1 MNP can enter the pore. Even if more MNPs could enter the pore it is unlikely that they would have a significant effect on the ionic current.

We note that the manuscript neglected to indicate the nanopore was reduced in thickness for the sensing experiments and this has been rectified on page 15, lines 1-2.

“For application here in quantitative analysis, the SiN membranes have been thinned down to about 80 nm.”

The relevant values of the thickness for employed SiN membranes in Figure 2, 3 and 4 have been noted now in the revised manuscript as well.

See page 7, line 7-8 in the figure legend for Figure 2

“... 180-nm-thick SiN membranes were employed here. ...”

See page 8, line 16 in the figure legend for Figure 3

“... on a 180-nm-thick SiN membrane ...”

See page 11, lines 4-5 in the figure legend for Figure 4

“... SiN membranes of a thickness of 180 nm were employed here for the paralleled detection and simulation work.”

Comment 6: If possible, please have SEM images or TEM images to show the MNPs are in the nanopores. □

Authors' response: We've added a SEM image showing the presence of a magnetic nanoparticle sitting inside a nanopore in the Supplementary Information (Supplementary Figure 11).

Supplementary Figure 11. SEM image of a MNP sitting inside a SiN nanopore. The image was taken on a FEI Nova NanoSEM 450 instrument operated in backscattered electron imaging mode. Scale bar, 100 nm.

The following text was added on page 6, lines 15-16

“A SEM image of a MNP inside the nanopore is shown in Supplementary Fig. 11.”

Comment 7: In line 269-270, “It is thus possible to detect large numbers of blocking and unblocking events simultaneously because most of the pores likely capture only one nanoparticle at a time”. Any evidence or reference? □

Authors’ response: We agree with the referee that this sentence implies more than we intended. What we were trying to say is that in an array of nanopores, more than one pore can be blocked in an array and this can be used for quantification. The nanopores used for the sensing experiments have been rationally engineered to hold one particle at a time. The other thing is that these nanopores have been placed far away from each other such that the nanopore electric fields would not interfere between each other and so nanopores from the fabricated nanopore array are working independently, ready for sensing experiments with even larger number of nanopores. We have amended this sentence on page 12, lines 13-15 to read

“It is thus possible to use arrays of nanopores in sensing to detect multiple blocking and unblocking events simultaneously which will facilitate in using nanopore blockade sensors for quantitative analysis.”

Comment 8: Please add more references of micro-nanodevices for detecting PSA. Compare the device fabrication advantages and disadvantages with them. As we know EBL is quite expensive to fabricate nanopores. In addition, is it necessary to detect PSA at sub-fM concentration for real medical practice? □

(1) Damborska, D., Bertok, T., Dosekova, E., Holazova, A., Lorencova, L., Kasak, P., & Tkac, J. (2017). Nanomaterial-based biosensors for detection of prostate specific antigen. *Microchimica Acta*, 184(9), 3049-3067. □

(2) Stern, E., Vacic, A., Rajan, N. K., Criscione, J. M., Park, J., Ilic, B. R., ... & Fahmy, T. M. (2010). Label-free biomarker detection from whole blood. *Nature nanotechnology*, 5(2), 138. □

(3) Alzghoul, S., Hailat, M., Zivanovic, S., Que, L., & Shah, G. V. (2016). Measurement of serum prostate cancer markers using a nanopore thin film based optofluidic chip. *Biosensors and Bioelectronics*, 77, 491-498. □

(4) Kosaka, P. M., Pini, V., Ruz, J. J., Da Silva, R. A., González, M. U., Ramos, D., ... & Tamayo, J. (2014). Detection of cancer biomarkers in serum using a hybrid mechanical and optoplasmonic nanosensor. *Nature nanotechnology*, 9(12), 1047. □

(5) Zheng, G., Patolsky, F., Cui, Y., Wang, W. U., & Lieber, C. M. (2005). Multiplexed electrical detection of cancer markers with nanowire sensor arrays. *Nature biotechnology*, 23(10), 1294. □

Authors' response: These reference papers have been added now in the revised manuscript.

Please note that in this case the PSA was a model analyte, as indicated on page 3, line 20. There are in fact many analytes that require such low detection limits. This was discussed in a paper reported by Rissin et al. (Rissin, D. M.; Kan, C. W.; Campbell, T. G. et al. Single-molecule enzyme-linked immunosorbent assay detects serum proteins at subfemtomolar concentrations. *Nature Biotechnology* **2010**, 28, 595-599.), a typical concentration level in serum from a 1-mm³ tumor is about 2 fM, which is far below the detection limit for conventional ELISA assays. It was also discussed in depth by Mirkin and co-workers (Giljohann, D. A.; Mirkin, C. A., Drivers of biodiagnostic development. *Nature* **2009**, 462, 461-464) which stated that ultrasensitive sensors were needed for both diagnosis and treatment efficacy. In fact PSA is specifically mentioned in this latter paper as an important analyte to detect at ultralow levels as after treatment of prostate

cancer, patients need to wait two years to find out whether the treatment was effective or not. This time frame is dictated by the sensitivity of existing PSA measurement methods and more sensitive technologies would reduce this time and the associated stress significantly. We have refrained from discussing this in the paper as we wanted the emphasis of the paper to be on the generic-ness of the concept rather than as a sensor for a given species.

The following text has been added on page 3, lines 21-24

“Regardless of PSA being a model analyte here that could be changed to any other protein for which there are suitable antibodies, the case for detecting PSA at low concentration is for better treatment efficacy.²⁰ As such there has been a number of microdevices developed for detecting ultralow levels of PSA.²¹⁻²⁵”

Comment 9: The authors use the number of blockage events to assess the detection limit of the nanopore sensors. This is very interesting. Why amplitude of ionic current change is not used to estimate the detection limit? Based on authors’ statement, ideally one nanopore can capture one MNPs, leading to one blockage event. Is it possible that one nanopore can capture two or more MNPs? If so, how about the accuracy to estimate the detection limit? □

Authors’ response: The idea of using the steps in current as a result of blockage events instead of using the value of decreased ionic currents for quantification of PSA concentrations is to ensure the analytical signal is digital and robust. That is the nanopore is either in the blocked or unblocked state such that small variations in size during fabrication has no effect.

The size of nanopores in the nanopore blockade sensors reported in this study is designed to be able to have one nanoparticle at most on each time. Based on our experience and knowledge, we do not think it is possible to have more than one nanoparticle inside a nanopore. The following text was added to the discussion on page 15, lines 19-22

“The blockages give robust analytical signals where large changes in current are observed as each nanopore is blocked such that small variations in pore size during fabrication, or proteins translocating through the nanopores without an associated magnetic nanoparticle do not give false signals.”

Comment 10: More explanations of Figure 5.c are needed in the text. □

Authors' response: Additional discussion has been added on page 13, lines 11-17.

“A correlation between the number of detected irreversible blockades and PSA levels had been performed. A monotonic trend was found that when a higher concentration of PSA was present, the mean number of detected irreversible blockades increased as well. Interestingly, there was no irreversible blockade detected when there was 0.4 fM PSA present in the electrolyte solution. Afterwards, the nanopore blockade sensors were challenged with plasma samples that had been diluted to have even lower concentration of PSA.”

Answers to referee #2:

Comment 1: The authors should discuss the specificity of epitopes 1 and 5 chosen for targeting PSA and if there's any known nonspecific binding of the antibodies they used to target these specific epitopes. The authors should also discuss or quantify the number of antibody sites per nanopore and per MNP.

Authors' response: Thank you for your comment. Additional discussion has been added in the revised manuscript.

In terms of the design strategy to construct a sandwich immunocomplex structure with two sandwiched anti-PSA antibodies, in this study the anti-PSA antibody (ab10187) and the anti-PSA antibody (ab10185) were employed as the detection antibody and capture antibody against PSA and had been immobilised onto the surfaces of nanopores and MNPs, respectively. These are commercial antibodies with the target epitopes on PSA molecule by these two anti-PSA antibodies being different. Importantly the information from the supplier Abcam suggests there is no cross reactivity with human kallikrein 2 which is also found in the prostate and has similarities with PSA. Generally the epitopes on a PSA molecule can be classified into two groups, i.e., the hidden epitopes and the exposed epitopes, making them to be targeted for the detection of free PSA or total PSA including complexed ones (*Biosensors & Bioelectronics* **2009**, 24, 2678-2683.). In this study, the capture antibody (ab10185) binds to the exposed epitope (epitope 5) whilst the detection antibody (ab10187) associates with the hidden epitope (epitope 1), which makes this nanopore blockade sensor is specifically detecting free PSA in our study. The choice of employing these two anti-PSA antibodies as the capture antibody and detection antibody actually follows the experience and knowledge available from literature papers as well as the manufacturer's recommendation (*Journal of Clinical Laboratory Analysis* **2011**, 25, 37-42.; *Chemical Papers* **2015**, 69, 143-149.). It is true that the nonspecific binding from some substance in serum, e.g., the immunoglobulins of the M class (IgM), may interfere by binding to anti-PSA antibodies and results in high background intensity (*Human Immunology* **2009**, 70, 496-501.). However, different from conventional solid phase based assays, nanopore blockade sensors here introduced the concept of magnetic analyte shuffling that differentiates between specific binding and nonspecific binding by using magnetophoretic force. This process helps to exclude nonspecific binding and thus enhance the specificity for nanopore blockade sensors, by using magnetic field reversal to simply pull out these magnetic nanoparticles with

unbound specific analyte molecules. To ensure the designing on nanopore blockade sensors with these two anti-PSA antibodies is clear, the following sentence has been added on page 18, lines 24-27 to read

“Antibody specificity. The anti-PSA antibodies used in this study were the mouse IgG monoclonal anti-PSA specific antibodies that one anti-PSA antibody (ab10187) is specific for epitope 1 (only free PSA) while the other anti-PSA antibody (ab10185) is specific for epitope 5 of PSA (both free PSA and complexed PSA).”

Regarding the number of immobilised anti-PSA antibodies on a MNP, this is very challenging to unambiguously determine. A simplified calculation with an assumption about the cross-section molecular size of 8.5 nm × 4.0 nm for an anti-PSA antibody and a 50% coupling efficiency for the attachment of anti-PSA antibodies onto it. Note that typical molecular dimensions for IgG protein are approximately 14.5 nm × 8.5 nm × 4.0 nm (*ACS Nano* **2008**, 2, 2374-2384). Also the molecular mass is around 150~170 KDa depending on a species. Here we calculate the possible number of attached anti-PSA antibodies on a MNP. Take a typical cubic MNP of 50 nm in length as an example, the total surface area for the cube would be $1.5 \times 10^4 \text{ nm}^2$ ($= 6 \times 50 \text{ nm} \times 50 \text{ nm}$). So the possible maximum coverage (50%) for anti-PSA antibodies on it would be $7.5 \times 10^3 \text{ nm}^2$. Therefore, the estimated number of anti-PSA antibodies would be about 220 ($= 7.5 \times 10^3 \text{ nm}^2 / (8.5 \text{ nm} \times 4.0 \text{ nm})$). And so is the similar calculation for a functionalised nanopore. The interior surface area for a conical nanopore is about 15764 nm². So the estimated number for anti-PSA antibodies would be about 231. (The nanopore interior surface area can be calculated from the equation $\pi(r+R)S = \pi(r+R)\sqrt{[(R-r)^2+h^2]}$, where r and R stand for the radius of the bottom and top pore orifice (30 nm and 88 nm), respectively. And h stands for the membrane thickness (80 nm). Note that the above calculations are obtained based on simplified assumptions. The real numbers of anti-PSA antibodies on the surfaces of a nanopore interior and a MNP could be differed somehow, especially when there exist more factors that could influence the immobilisation process.

We feel the inclusion of such estimates might detract from the central concept of the paper. As such we have refrained from adding this to the text of the manuscript. We will happily add this to the paper if the referee feels it is an imperative.

Comment 2: However, this currently reported method still would have the issue with

larger biomolecules that passively pass by and block the nanopore. Although the authors do report that the electrophoretic charges would dispel the attraction of negatively charged molecules such as BSA and PSA and other proteins to the negatively charged nanopore surface. This was further validated when using a human blood sample where the plasma platelets were first isolated before incubation with the MNPs. Further discussion on how less negatively charged biomolecules or positively charged biomolecules could impact the technology. Some added discussion on what cases would the magnetic forces overcome the electrophoretic forces and vice versa and which would play the more critical role in the determination of the combined analyte + MNP being able to make it into the nanopore for complete blocking.

Authors' response: The nanopores are in fact much larger than even large biomolecules, being 30 nm in diameter at their smallest orifice. The translocation of the very largest biomolecules will in fact not affect the identification of a blockage event due to the way the measurement is performed. The identification of a blockage event is determined by a significantly larger change in current than a protein translocation event. The presence of proteins in the sample simply affects the background noise and so the charge of the protein and it translocating, only affects the background noise. This is now commented, on page 16, lines 5-10 with the text

“Performing the measurement in complex biological media does mean that other proteins within the sample could translocate through the pore but will not block it as the nanopore is too large, ~30 nm at its smallest orifice. As such, these translocation events may increase the background noise but will not be counted as a binding event as any change in current is not permanent as it is when the immuno-sandwich is formed.”

Note cells and platelets could also of course blocks the nanopores. One of the advantage so using magnetic nanoparticles is the capture of the analyte and the measurement of the blockages can be spatially and temporally separated such that the blood sample is not exposed to the nanopores, just the magnetic nanoparticles for the capture of the protein analyte.

Comment 3: The size of nanopores were specifically customized to allow for the MNP to pass through but not translocate through the nanopore. Using a patch clamp setup, they were able to measure the number of nanopores that were now blocked to the quantify

the number of PSA molecules detected. However, the authors also discussed the potential impact of how the MNP traveled into the nanopore could impact the ability for PSA+MNP to block the nanopore properly. It would be helpful for the authors to have added discussion about how to design and optimize the pattern of nanopores for the biological sample (i.e. how far apart or the minimum distance between the nanopores to still maintain the sensitivity of molecule detection – is there a distance small enough at which a PSA+MNP becomes stuck between two open nanopores and unable to make its way into one of them).

Authors' response: The nanopore arrays are designed such that the nanopores are spaced far enough apart such that the electric field at each nanopore will not affect adjacent nanopores. In such an arrangement each nanopore in the nanopore array contributes a similar amount of ionic current to the total measured sum of ionic current when all the nanopores on a membrane are in the opened state. According to theoretical papers (Sub-additive ionic transport across arrays of solid-state nanopores. *Physics of Fluids* **2014**, 26, 012005.; Crosstalk between adjacent nanopores in a solid-state membrane array for multianalyte high-throughput biomolecule detection. *Journal of Applied Physics* **2016**, 120, 064701), a distance of 5 μm is far enough for electric fields from two neighboring 30-nm nanopores to not interfere with each other. The sensitivity for detecting individual blockade events is not limited by the pore distance, but the amplitude of decreased ionic current. However, the magnitude of pore-blocking induced current reductions is within few hundreds of pA, which is totally detectable and high enough for a common patch clamp to measure it.

Currently, in our group we fix the pore spacing to be 5 μm . According to our simulation results in Figure 3, if a nanoparticle lands onto a place where it is more than 250 nm far away from a nanopore, it would not be affected by the localised electric field from the nanopore and thus unable to make its way to this nanopore.

The following sentences have been added into the revised manuscript for additional discussion on page 11, lines 11-15

“Here nanopore arrays fabricated on SiN membranes have been rationally placed far away enough between adjacent nanopores such that the adjacent nanopore electric field would not interfere each other and thus nanopores work independently and contribute to similar amount of ionic current.”

Comment 4: Similarly, although this appears to be a great tool for low levels of detection of PSA. The authors should discuss how these new concentration levels compare to the physiological levels of PSA, and what levels are important for diagnostic purposes such as in prostate cancer or benign prostatic hyperplasia (BPH). As a biomarker, PSA already provides a significant number of false positives and is primarily used as a trending biomarker and not a diagnostic biomarker (i.e. knowing that we have a very low level of PSA is not particularly meaningful and knowing the rate at which the PSA levels have changed is more diagnostically important).

Authors' response: As discussed above there are in fact many analytes that require such low detection limits and in this case PSA was used as a model analyte. Rissin et al. (*Nature Biotechnology* **2010**, 28, 595-599.), discussed the typical concentration level in serum from a 1 mm³ tumor is about 2 fM, which is far below the detection limit for conventional ELISA assays, hence the unmet need. Even with PSA, although a controversial biomarker for diagnosis, it is an important biomarker for treatment efficacy as discussed in depth by Mirkin and co-workers (Giljohann, D. A.; Mirkin, C. A., Drivers of biodiagnostic development. *Nature* **2009**, 462, 461-464). In this case ultrasensitive sensors for PSA are needed as after treatment of prostate cancer, patients need to wait two years to find out whether the treatment was effective. This time frame is dictated by the sensitivity of existing PSA measurement methods and more sensitive technologies would reduce this time and the associated stress significantly. As indicated for referee #1 above the following text has been added on page 3, lines 21-24

“Regardless of PSA being a model analyte here that could be changed to any other protein for which there are suitable antibodies, the case for detecting PSA at low concentration is for better treatment efficacy.²⁰ As such there has been a number of microdevices developed for detecting ultralow levels of PSA.²¹⁻²⁵”

Comment 5: This new report certainly provides a new leap forward in bionanosensing using nanopores by moving from passive detection to active detection. However, the technology's specificity becomes limited to the specificity of the antibody to the epitopes of the PSA. Nonspecific binding by these antibodies could potentially contribute to mis-capturing and counting. Some further discussion on this would be helpful. Overall, this is a well written paper with good mix between the theoretical / computational aspects and

the experimental aspects of nanopores, presents a new view in biosensing using nanopores, and provides an initial good proof of principle demonstration of the reported platform technology.

Authors' response: It is true for conventional solid phase based assays that the nonspecific binding from some substance in serum, e.g., the immunoglobulins of the M class (IgM), may interfere by binding to anti-PSA antibodies and results in high background signal (*Human Immunology* **2009**, 70, 496-501.). However, different from the conventional solid phase based assays, nanopore blockade sensors here introduced the concept of magnetic analyte shuffling that can differentiate between specific binding and nonspecific binding by using magnetophoretic force. The binding affinity from a nonspecific molecule to an anti-PSA antibody would be significantly different, compared to that of a specific pairing affinity quantified recently from measurements with atomic force microscopy at single molecule level (56 ± 2 pN, *Analytical Chemistry* **2010**, 82, 5189-5194.). Therefore, this magnetic analyte shuttling process in the nanopore blockade sensors helps to exclude nonspecific binding and false positives by using magnetic field reversal to simply pull out these magnetic nanoparticles with unbound specific analyte molecules.

The following sentence has been added in the revised manuscript on page 16, lines 1-3 to read

“That is nanoparticles that block the pores nonspecifically are removed by reversing the magnetic field while if the immuno-sandwich is formed the magnetic nanoparticles cannot be removed.”

Answers to referee #3:

Comment 1: while the author presents data that they can detect an antigen target at low concentration, it is not clear that this system can accurately distinguish different concentrations of target from a given sample or how reliable these measurements would be in a panel of human samples. Specifically, Figure 5C indicates that there is considerable overlap between the values obtained at each concentration and there appears to be a very limited detection range. This data is not adequately discussed in the text or figure legends. Does this data represent the mean of the samples? If so, how many samples were analyzed at each point? Do the error bars indicate standard deviations or the standard error of the mean? Were there significant differences between the values at each concentration?

Authors' response: Thank you for the comment. The text and/figure legends have been properly revised and detailed interpretation and discussion have been added on page 13, lines 11-17.

“A correlation between the number of detected irreversible blockades and PSA levels had been performed. A monotonic trend was found that when a higher concentration of PSA was present, the mean number of detected irreversible blockades increased as well. Interestingly, there was no irreversible blockade detected when there was 0.4 fM PSA present in the electrolyte solution. Afterwards, the nanopore blockade sensors were challenged with plasma samples that had been diluted to have even lower concentration of PSA.”

On each concentration, at least five parallel measurements from different chips were performed and the error bars stand for the standard deviations between these five measurements. The current nanopore blockade sensors with 3×3 pore array, already can response sensitively to PSA of different concentrations. It has been demonstrated from determination of PSA levels on plasma samples, showing comparable detection performance to ELISA kit while the plasma samples have been diluted to have much lower concentrations. We anticipate that with more nanopores present on the SiN membrane, our nanopore blockade sensors can reach to a response to analytes in a larger dynamic range with higher confidence and narrowed deviations for measurements.

Comment 2: There is also not sufficient experimental to evaluate the utility of method. Nanoparticles are drawn to the membrane with a magnetic field that is cycled four times to allow specific enrichment of the magnetic nanoparticles that have bound the target protein, while the antigen-free nanoparticles are removed when the magnetic field is inverted to draw the particles away from the membrane. However, the authors do not describe whether the nanopore membrane can be regenerated by application of a stronger magnetic field or other method to allow reanalysis of the sample. Repeated measurements are needed to evaluate the accuracy and reproducibility of the assay results.

Authors' response: About the regeneration of used chips after ionic current experiments, actually the chips employed for experiments throughout this study include these regenerated chips as well. A brief description about the regeneration procedure is provided as follows. After ionic current experiments, the chips have been washed thoroughly by electrolyte solution and deionized water. The magnetic nanoparticle in a nanopore could be pulled out by using a neodymium magnet with strong magnetic field and the state for a nanopore (blocked or unblocked) could be examined by a dark-field optical microscopy. Once it is confirmed from the optical examination that all the 3×3 nanopores are opened, these chips then continue to be renewed by a 10-min oxygen plasma treatment on both sides, respectively and re-modified to have functional interfaces with immobilised anti-PSA antibodies for further measurement experiments. Prior to be employed for collecting data for a typical ionic current experiment with a regenerated chip, the analysis of resistance and noise level from this chip is carefully examined to be sure that it is in a similar condition, compared to these non-renewed chips modified with anti-PSA antibodies.

According to our experience, after multiple times for chip regeneration, these renewed chips could have enlarged nanopores with a larger pore diameter and thus a lower pore resistance, which means that these chips are no longer useful for the nanopore blockade sensors. Nonetheless, before the pore diameter increases to be big enough that unexpected passage for magnetic nanoparticles occurs, these chips still can function normally, comparably and similarly as these non-renewed chips for experiments including resistance measurement and ionic current experiment employed for sensing work.

This reported nanopore blockade sensors give robust analytical signals for quantitative analysis by working with the numbers of stepwise drops in ionic current, but not the

magnitude of ionic current reductions. As such the small changes in pore diameter during fabrication, regeneration or between different chips do not affect the robustness and detection performance.

We've added the following sentences in the revised manuscript between page 20, lines 24-28 and page 21, lines 1-3

“After ionic current experiment, these used chips can be regenerated for multiple usages. The nanoparticle induced pore blockades can be unblocked by applying an external strong magnet to draw the MNPs out of the nanopore. After dark-field optical microscopy is used to confirm that the nanopores are in opened state, the chips were cleaned with 10 minutes in an oxygen plasma to both sides of the chip. The chip can then be re-modified using the same procedure as described in the chip modification section of the paper. Prior to be used, a careful examination on the measured resistance and noise level was performed to ensure these regenerated chips behaviour similarly and comparably as the non-renewed chips.”

Reviewer #1 (Remarks to the Author):

The authors have addressed the reviewer's comments adequately. Hence, the reviewer recommends acceptance of this article..

Reviewer #2 (Remarks to the Author):

Overall the authors have adequately addressed most of the comments provided in the initial review. I would strongly encourage for the information in the reply to comment #1 of my initial review to be included into the supplemental information if the authors feel strongly about not including it into the main text of the article. This is in regards to the theoretical calculation of how many potential binding sites there are in the nano pore. As a follow up, it was mentioned that there are potentially 220-231 potential binding sites within the nano pore. Within that case, how would the application of the magnetic field change subsequent results If there was more than one antibody binding the PSA. Could there be a strong enough magnetic field that would allow for single 1:1 binding events to dissociate as opposed to binding events where 2+ antibodies bind to the PSA.

On the previous comment #3 I posted, the authors should specify the minimum spacing as provided by other reports and provided the citation in the main text of the paper itself and not only in the responses.

In regards to the previous nonspecific binding comments, the authors need to address that there are going to be instances where the antibody will bind with similar affinity to non-PSA entities (although it might happen on a much a smaller time scale than the MNP-PSA tagged entities). These types of nonspecific binding events would not result in the these non-PSA entities to be removed under changes in the magnetic field. More specifically, I am referring to non-specific binding to the anti-PSA that is in the nano pore not the anti-PSA that is attached to the MNP (which I understand would be removed out of the nanopore). Hence, the authors do need to state that the limitations of this technology will be partially limited to the binding specificity of the used antibodies.

In addition, changes in the actual text of the paper needs to incorporate the provided citations as to why epitopes 1 and 5 were specifically chosen for the experiments described in this paper (as provided by the authors - Journal of Clinical Laboratory Analysis 2011, 25, 37-42.; Chemical Papers 2015, 69, 143-149.)

The authors should also described in the main article what "whole blood" refers to - is this human or animal blood (i.e. what species)? If human, describe where it was provided from or collected by, include statement re: IRB approval if human samples. If from animal, similar described where it came from, how it was acquired, and include statement regarding IACUC approvals.

Reviewer #3 (Remarks to the Author):

My original review of the manuscript by Chuah et al. indicated that there were several shortcomings with the study. Unfortunately, these problems have not been addressed by the current revision.

A major problem mentioned in the first review was the lack of description and statistical analysis for the limited sample quantitation data. The data and the analysis presented in still not sufficient. The added text still does not indicate the number of samples or replicates analyzed in Fig. 5c-d, describe the format of the data, or explain how they were analyzed. In Figure 5c, the authors indicate that there is a monotonic trend for greater nanopore blockade with increasing sample concentration. However, for this method to be useful, the nanopore assay results would need to be able to distinguish and quantitate different biomarker concentrations. The data presented indicate that the current method cannot do this. The data depicted in Figure 5d is also not correctly presented to support the contention that there is a correlation between the nanopore and ELISA

data, which does not appear reasonable given the variation present in the nanopore data shown. The response also indicates that the sensors employed in this manuscript are essentially single-use, since the description indicates that the chips would need to be examined by dark-field optical microscopy to confirm their pores are not still blocked, renewed by oxygen plasma treatment, and conjugated with capture antibodies and a blocking agent before they would be suitable for reuse. The authors indicate that these chips would also have to be analyzed for changes in resistance and noise, due to the erosion of the pores during this process. This would likely increase the variation of any repeat measurements made with the same chip. Sample replicates would therefore need to be analyzed using separate chips to conduct analyses in a reasonable timeframe, which might be possible if the system could be multiplexed and made quantitative.

At present, however, it is not clear how this approach could be applied to generate quantitative data. It appears that significant advances in chip performance and assay multiplexing would need to occur before this method could be used to generate reliable data.

Responses to reviewers' comments

Reviewer #2 (Remarks to the Author):

Comment 1: Overall the authors have adequately addressed most of the comments provided in the initial review. I would strongly encourage for the information in the reply to comment #1 of my initial review to be included into the supplemental information if the authors feel strongly about not including it into the main text of the article. This is in regards to the theoretical calculation of how many potential binding sites there are in the nanopore.

Authors' response: This is an excellent recommendation. As suggested by the referee, the following text has been added into the Supplementary Information on page 15.

“Section S5: Estimate of the number of anti-PSA antibody sites on a single MNP as well as a single nanopore

It is very challenging to unambiguously determine the exact number of immobilised anti-PSA antibodies on a single MNP as well as a single nanopore. Here, the estimation was made by calculations based on simplified assumptions.

A simplified calculation was made with an assumption about the cross-section molecular size of 8.5 nm × 4.0 nm for an anti-PSA antibody and a 50% coupling efficiency for the attachment of anti-PSA antibodies onto it. Note that typical molecular dimensions for IgG protein are approximately 14.5 nm × 8.5 nm × 4.0 nm.¹¹ Also the molecular mass is around 150~170 KDa depending on a species. Here we calculate the possible number of attached anti-PSA antibodies on a MNP. Take a typical cubic MNP of 50 nm in length as an example, the total surface area for the cube would be $1.5 \times 10^4 \text{ nm}^2$ ($= 6 \times 50 \text{ nm} \times 50 \text{ nm}$). So the possible maximum coverage (50%) for anti-PSA antibodies on it would be $7.5 \times 10^3 \text{ nm}^2$. Therefore, the estimated number of anti-PSA antibodies would be about 220 ($= 7.5 \times 10^3 \text{ nm}^2 / (8.5 \text{ nm} \times 4.0 \text{ nm})$).

And so is the similar calculation for the functionalised nanopore. The interior surface area for a conical nanopore is about 15764 nm². So the estimated number for anti-PSA antibodies would be about 231. The nanopore interior surface area is calculated from the equation $\pi(r+R)S = \pi(r+R)\sqrt{[(R-r)^2+h^2]}$, where r and R stand for the radius of the bottom and top pore orifice (30 nm and 88 nm), respectively. And h stands for the membrane thickness (80 nm).”

Comment 2: As a follow up, it was mentioned that there are potentially 220-231 potential binding sites within the nanopore. Within that case, how would the application of the magnetic field change subsequent results If there was more than one antibody binding the PSA. Could there be a strong enough magnetic field that would allow for single 1:1 binding events to dissociate as opposed to binding events where 2+ antibodies bind to the PSA.

Authors' response: This is a very interesting query. According to Poisson statistics, it is possible to have a single functionalised MNP bound with two PSA molecules, but with extremely low probability. As discussed in the main manuscript, for sensing 0.8 fM PSA, it is likely that only 18 out of 1.87×10^9 (anti-PSA)-MNPs would have captured more than one PSA molecule. However, as the antibodies are monoclonal it

is hard for us to envisage a scenario where more than one antibody could bind to the same epitope.

In regard to the capability of differentiating the number of unbound single-molecule antigen-antibody events between the MNP and the nanopore during unblocking process, it seems possible to having a magnetic force applied on a single MNP to discriminate events with different binding strength by tuning the strength of magnetic field, e.g., adjusting the applied voltage on the switchable electromagnet and the distance between the MNP and the magnet. Other factors may need to take into account as well in this scenario including temperature, salt concentration and possible nanoconfinement within the pore. This is something we are considering exploring in later studies but feel at this stage we have the ability to discriminate between nonspecific entry to the nanopore and one PSA molecule being captured which we regard as a significant advance.

Comment 3: On the previous comment #3 I posted, the authors should specify the minimum spacing as provided by other reports and provided the citation in the main text of the paper itself and not only in the responses.

Authors' response: Based on these previous studies about the crosstalk between adjacent nanopores and the global conductance of N pores in a 2D square-lattice array, the minimum spacing between nanopores in an array of 3×3 nanopores to avoid crosstalk is about $0.5 \mu\text{m}$ in our case.

The following text along with this value of the spacing ($0.5 \mu\text{m}$) has been added into the main manuscript with the citations of these previous theoretical papers (reference number 31 and 32) on page 11, lines 14-17

“Here nanopore arrays fabricated on SiN membranes have been rationally placed far away enough ($\gg 0.5 \mu\text{m}$) between adjacent nanopores such that the adjacent nanopore electric field would not interfere each other and thus nanopores work independently and contribute to similar amount of ionic current^{31, 32}.”

Comment 4: In regards to the previous nonspecific binding comments, the authors need to address that there are going to be instances where the antibody will bind with similar affinity to non-PSA entities (although it might happen on a much a smaller time scale than the MNP-PSA tagged entities). These types of nonspecific binding events would not result in these non-PSA entities to be removed under changes in the magnetic field. More specifically, I am referring to non-specific binding to the anti-PSA that is in the nanopore not the anti-PSA that is attached to the MNP (which I understand would be removed out of the nanopore). Hence, the authors do need to state that the limitations of this technology will be partially limited to the binding specificity of the used antibodies.

Authors' response: This is a very perceptive comment from the reviewer and something we worried about. To avoid such a possibility within the experiments presented in the paper, the sample solutions were not directly introduced into the electrolyte solution to perform ionic current experiments. It is the functionalised MNPs that are dispersed into sample, collected by a magnet and washed before its introduction into the electrolyte solution for sensing measurements. Therefore, it is unlikely that the interferences from samples could come into the functionalised nanopores. But we do admit that the utilisation of this developed nanopore sensor could potentially and partially be limited by the available choice of analyte-specific

receptors (antibodies in this work). It is also worth adding that even if there is nonspecific binding to the antibodies within the nanopore that will not result in a major blockage of the nanopore and therefore it will not be observed as a significant current change. Furthermore, as the nanopore has many available antibodies, we think it is unlikely that such nonspecific binding, when rare, will prevent PSA captured by the nanoparticles from binding to the nanopore.

To clearly clarify the limitation of the nanopore blockade sensors, the following text about the specificity of the antibodies, is now added to the main text on page 16, lines 26-28, and page 17, lines 1-4.

“One potential concern for nanopore blockade sensors is if other species bind to the antibodies within the nanopores. It seems plausible that such binding events may decrease the ionic current. However, with the nanopores being 30 nm in diameter at its minimum, a single protein binding event is unlikely to cause a current drop of the magnitude associated with the nanoparticle binding to the nanopore. In the nanopore blockade sensor format presented herein, the capture of the analyte is by the (anti-PSA)-MNPs and the nanopore measurement are spatially separated.”

Comment 5: In addition, changes in the actual text of the paper needs to incorporate the provided citations as to why epitopes 1 and 5 were specifically chosen for the experiments described in this paper (as provided by the authors - Journal of Clinical Laboratory Analysis 2011, 25, 37-42.; Chemical Papers 2015, 69, 143-149.)

Authors' response: These two papers are now cited in the methods section of the paper.

“**Antibody specificity.** The anti-PSA antibodies used in this study were the mouse IgG monoclonal anti-PSA specific antibodies that one anti-PSA antibody (ab10187) is specific for epitope 1 (only free PSA) while the other anti-PSA antibody (ab10185) is specific for epitope 5 of PSA (both free PSA and complexed PSA).^{37, 38}”

Comment 6: The authors should also described in the main article what "whole blood" refers to - is this human or animal blood (i.e. what species)? If human, describe where it was provided from or collected by, include statement re: IRB approval if human samples. If from animal, similar described where it came from, how it was acquired, and include statement regarding IACUC approvals.

Authors' response: The human whole blood samples used in this work is described in the methods part in the main text, which is on page 17.

It says, “Human whole blood (anticoagulated with K2 EDTA) was purchased from Innovative Research (Novi, USA).”

Reviewer #3 (Remarks to the Author):

Comment 1: A major problem mentioned in the first review was the lack of description and statistical analysis for the limited sample quantitation data. The data and the analysis presented in still not sufficient. The added text still does not indicate the number of samples or replicates analyzed in Fig. 5c-d, describe the format of the data, or explain how they were analyzed. In Figure 5c, the authors indicate that there

is a monotonic trend for greater nanopore blockade with increasing sample concentration. However, for this method to be useful, the nanopore assay results would need to be able to distinguish and quantitate different biomarker concentrations. The data presented indicate that the current method cannot do this. The data depicted in Figure 5d is also not correctly presented to support the contention that there is a correlation between the nanopore and ELISA data, which does not appear reasonable given the variation present in the nanopore data shown.

Authors' response: We apologise for placing the wrong emphasis on the previous revision in relation to this referee's points and appreciate how they have made the paper better. We have repeated a significant number of experiments to refine the calibration curve and give it great statistical validity. The calibration curve (Figure 5c in the main manuscript) is now revised and is attached below in Figure S1. This revised calibration curve is plotted by the correlation between the determined value of mean specific blockades and different PSA concentrations. The uncertainties presented herein were calculated using Poisson statistics. The exact method was used to calculate the 95% confidence intervals of the mean for Poisson distribution, which shows how precisely the averaged specific blockades have been determined at each concentration. A close correlation between the determined mean blockades and the PSA concentration is present in the Figure. Repeated measurements from separate nanopore chips were performed at each PSA concentration in this figure. The ranges of 95% confidence intervals in Figure S1 are reasonably good for just 9 nanopores. In addition to this, we have calculated the p value between any two values of the determined mean specific blockades, and labelled the results in the calibration curve to show the significance for distinguishing two concentrations of PSA. As depicted in the Figure, there is a clear statistical difference between the blank and the detection limit of 0.8 fM and although the uncertainties obtained from 9 nanopores mean there is not a significant difference between say 4 and 8 fM there is for larger changes in concentration. So we agree with the referee that the system can be improved but we feel that the concept is well realised in this paper. We believe that the strategy of magnetically capturing analyte to bring it the nanopore and to block the pore is a major advance in that we can detect low concentrations of species and perform analyses in complex biological fluids.

Figure S1. Irreversible blockades observed versus PSA concentrations. Blockade events were counted after 4 cycles of switching on the *trans*-magnet for 10 minutes, and subsequently the *cis*-magnet for 5 minutes to maximize probability of blockade events at extremely low concentrations. Mean value of irreversible blockades as well

as 95% exact confidence intervals were obtained from a total of 31 measurements (one measurement per chip). Of these measurements, 23 chips were employed and 8 of them were renewed. The exact method was used to compute 95% confidence intervals for Poisson means. The difference between any two determined Poisson means was tested; ns not significant, * $p \leq 0.05$, ** $p \leq 0.01$, *** $p \leq 0.001$ and **** $p \leq 0.0001$.

Figure S2. The mean value and 95% exact confidence interval (red) upon the number of independent measurements for sensing 0.8 fM PSA. 7 nanopore chips were employed and 3 of them were regenerated later, contributing to a total of 10 separate measurements of 0.8 fM PSA. Performing more measurements helps to obtain narrower 95% exact confidence intervals.

Also we have studied the effect of number of measurements on the precision of the obtained quantitation results by performing more measurements at samples of 0.8 fM PSA. It can be seen from the Figure S2 that the mean specific blockades stabilised at about 2.8 while the range of the 95% confidence interval is related to the number of measurements repeated. A narrower range of the determined value of the mean number of blockades at 95% confidence was received along with increasing the number of measurements, which indicates a greater precision.

Additionally, according to the referee's comment, Figure 5d for the comparison of PSA sensing from human whole blood samples by nanopore blockade sensors and a commercial ELISA kit was revised and attached below as well (Figure S3). The quantitation results from nanopore sensing were obtained from measurements by 6 separate nanopore chips, showing a closely matched sensing results with that from the ELISA kit.

Figure S3. Comparison of PSA level from whole blood using nanopore blockade sensor versus an ELISA kit. Error bars represent uncertainty of determined PSA concentration at 95% confidence by nanopore sensing at 6 chips (one measurement per chip) and the ELISA kit, respectively.

Comment 2: The response also indicates that the sensors employed in this manuscript are essentially single-use, since the description indicates that the chips would need to be examined by dark-field optical microscopy to confirm their pores are not still blocked, renewed by oxygen plasma treatment, and conjugated with capture antibodies and a blocking agent before they would be suitable for reuse. The authors indicate that these chips would also have to be analyzed for changes in resistance and noise, due to the erosion of the pores during this process. This would likely increase the variation of any repeat measurements made with the same chip. Sample replicates would therefore need to be analyzed using separate chips to conduct analyses in a reasonable timeframe, which might be possible if the system could be multiplexed and made quantitative.

Authors' response: This is an interesting comment. The nanopore chips utilised in this entire study were recyclable, i.e., renewed and repeatedly applied to sensing work following the procedures described in the methods part. Note that these careful examinations are made to ensure that thin substrate membranes (a thickness of ~80 nm) are not cracked or broken and the nanopores are ready for surface functionalisation and quantitation experiments. Note though that this was for experimental ease and in a real device the intention would be for the chips to be used only once.

Each value of the determined mean blockades from the calibration curve was constructed by averaging the measured specific blockades from different nanopore chips. Also the narrow 95% exact confidence intervals clearly indicate a great precision of measurements among these separate chips.

Comment 3: At present, however, it is not clear how this approach could be applied to generate quantitative data. It appears that significant advances in chip performance and assay multiplexing would need to occur before this method could be used to generate reliable data.

Authors' response: We feel this proof-of-concept study is a reasonably good start regarding how nanopore sensing for quantitative analysis based on the nanopore blockade sensors can be used. There are no doubts improvements can be made which we are currently doing for a follow-up paper. The established calibration curve presented in Figure S1 demonstrates this nanopore blockade sensors can generate quantitative signals upon responding to different concentrations of targeted PSA molecules. Furthermore, this quantitative capability was confirmed by the comparison of quantitation results between nanopore blockade sensors and the ELISA kit. In regard to multiplexing, we agree with the referee that it is one of the most important things for the nanopore sensing community to be expected in the near future.

Reviewer #1 (Remarks to the Author):

Comments have been addressed. Acceptance is recommended.